# Genome Wild Analysis and Molecular Understanding of the Aquaporin Diversity in Olive Trees (*Olea Europaea* L.)

**DOI:** 10.3390/ijms21114183

**Published:** 2020-06-11

**Authors:** Mohamed Faize, Boris Fumanal, Francisco Luque, Jorge A. Ramírez-Tejero, Zhi Zou, Xueying Qiao, Lydia Faize, Aurélie Gousset-Dupont, Patricia Roeckel-Drevet, Philippe Label, Jean-Stéphane Venisse

**Affiliations:** 1Laboratory of Plant Biotechnology, Ecology and Ecosystem Valorization, Faculty of Sciences, University Chouaib Doukkali, El Jadida 24000, Morocco; 2Université Clermont Auvergne, INRAE, PIAF, 63000 Clermont-Ferrand, France; boris.fumanal@uca.fr (B.F.); aurelie.gousset@uca.fr (A.G.-D.); patricia.drevet@uca.fr (P.R.-D.); philippe.label@inrae.fr (P.L.); 3Department of Experimental Biology, Center for Advanced Studies in Olive Grove and Olive Oils, University of Jaén, 23071 Jaén, Spain; fjluque@ujaen.es (F.L.); jrtejero@ujaen.es (J.A.R.-T.); 4Hainan Key Laboratory for Biosafety Monitoring and Molecular Breeding in Off-Season Reproduction Regions, Institute of Tropical Biosciences and Biotechnology, Chinese Academy of Tropical Agricultural Sciences, Haikou 571101, Hainan, China; zouzhi2008@126.com (Z.Z.); roslinda@lgm.gov.my (X.Q.); 5Group of Fruit Tree Biotechnology, Department of Plant Breeding, Murcia University, CEBAS CSIC, 30100 Murcia, Spain; lbremaud@cebas.csic.es

**Keywords:** aquaporin, *Olea europaea*, RNA-seq, functional diversity, *Oleaceae* evolution, plant domestication, *Verticillium dahliae*, cold and wound stress

## Abstract

Cellular aquaporin water channels (AQPs) constitute a large family of transmembrane proteins present throughout all kingdoms of life, playing important roles in the uptake of water and many solutes across the membranes. In olive trees, AQP diversity, protein features and their biological functions are still largely unknown. This study focuses on the structure and functional and evolution diversity of AQP subfamilies in two olive trees, the wild species *Olea europaea* var. *sylvestris* (*Oeu*AQPs) and the domesticated species *Olea europaea* cv. Picual (*Oleur*AQPs), and describes their involvement in different physiological processes of early plantlet development and in biotic and abiotic stress tolerance in the domesticated species. A scan of genomes from the wild and domesticated olive species revealed the presence of 52 and 79 genes encoding full-length AQP sequences, respectively. Cross-genera phylogenetic analysis with orthologous clustered *Olea*AQPs into five established subfamilies: PIP, TIP, NIP, SIP, and XIP. Subsequently, gene structures, protein motifs, substrate specificities and cellular localizations of the full length *Olea*AQPs were predicted. Functional prediction based on the NPA motif, ar/R selectivity filter, Froger’s and specificity-determining positions suggested differences in substrate specificities of Olea AQPs. Expression analysis of the *Oleur*AQP genes indicates that some genes are tissue-specific, whereas few others show differential expressions at different developmental stages and in response to various biotic and abiotic stresses. The current study presents the first detailed genome-wide analysis of the AQP gene family in olive trees and it provides valuable information for further functional analysis to infer the role of AQP in the adaptation of olive trees in diverse environmental conditions in order to help the genetic improvement of domesticated olive trees.

## 1. Introduction

Olive trees (*Olea europaea* L.) are among the most important crops around the Mediterranean basin. They are important not only as plant cultivation from a commercial point of view across the millennia, largely due to their multiple uses (e.g., oil, wood, ornamental uses, canned fruit, medicinal applications), but also to date, as a model to study oil fruit physiology. Its remarkable aptitude to face scarce precipitations and high summer temperatures enables it to grow in very diverse environments [1]. Most cultivars of interest show a high intraspecific genetic diversity, resulting, notably, in a wide range of contrasted stress responses. In this respect, identifying the molecular mechanisms underlying various stress tolerances in this species is essential.

Fundamentally, the effects of adverse growth conditions on plants vary among species. Most of the time, they lead to typical symptoms of tissue water imbalances. These deviant physiological situations trigger several highly specific and rapid hydraulic responses to readjust the internal water status (or the water homeostasis). Interestingly, the underlying molecular regulations have been mainly attributed to the action of aquaporins [2]. Indeed, several lines of data indicate that the abiotic and biotic stresses regulate the transcript accumulation, post-translational modifications and localization of various aquaporins, which correlate with the modulation of several ecophysiological traits such as osmotic potentials or hydraulic conductivities [3].

Aquaporins (AQPs) are proteins that are part of all biomembranes, i.e., the plasma membrane and the intracellular membrane system with the morphoplasm [4]. Whatever the hydraulic models and their levels of complexity (from the cell to the organism), AQPs take part in many physiological processes as key elements for the plant development and the cell responses to a plethora of environment stresses. To achieve this, AQP are players of prime importance in the regulation of hydraulic conductances, aided by their quick reaction time (seconds to minutes). In that respect, AQPs maintain the cell water homeostasis of cell turgor, by drawing water either from the apoplasm or from the vacuole (the main cell water capacitor), to buffer any changes in volume and osmolarity of the cytoplasm.

The typical AQP proteins consist of six trans-alpha helical transmembrane regions (TMH1-TMH6) connected by five loops (Loops A–Loops E, LA–LE). The carboxylic and amino terminals (*C*-t, and *N*-t, resp.) on either side of the primary sequence lie on the cytoplasmic side. The AQP protein monomer structure is highly conserved and resembles a tridimensional “hourglass model” [4]. Given the general conservation in such a 3D design, several amino acids along the primary sequence interact together in the folded structure, creating specific motifs that are important for the organization of constricts and then pore selectivity.

The first constrict consists of two additional half-helices (HB and HE), which form the seventh TM helix by the opposite LB and LE dipping into the membrane. Each half-helix possesses a highly conserved “asparagine–proline–alanine” (NPA) motif, forming a narrow region in the middle of the pore. The two asparagine side chains pointing into the pore are located at the end of these two half-helices, creating a dipole water movement and a proton (H^+^) exclusion environment. However, the general rule is that the size of this NPA constrict limits the size of the potentially transportable substrates [5]. The second constrict, known as “aromatic/arginine” or ar/R selectivity filter, consists of four residues (F58-H182-C191-R197 in AQP1). It is located towards the extracellular exit of the channel, approximately 8 Å away from the NPA regions. The ar/R constriction region forms the narrowest part of the pore, and it is therefore generally assumed that the discrepancies in its size and its hydrophobicity determine the solute transport specificity [6,7]. Similarly, five conserved Froger’s residues (P1–P5 residues, T116-S196-A200-F212-W213 in AQP1) have been found to play critical roles in regulating substrate specificity [6]. Moreover, it has been predicted that AQPs have nine specificity-determining positions (SDPs) for non-aqua substrates for each unique group [6]. All these motifs or positions of particular amino acids are viewed as key elements for the selection and the transport of permeants across the cell membranes.

From a tridimensional viewpoint, AQPs are arranged as homo- and/or heterotetrameric structures, which usually provide greater stability and folding of the proteins, resulting in highly efficient water transport across biological membranes [8]. A fifth channel forms through the middle of the tetramer array, which has been suggested to conduct gases, such as CO_2_ [9]. In addition to water, AQPs facilitate the transport of a variety of solutes, including boron, silicon, arsenite, ammonia, glycerol, polyols and urea, some ions (nitrate, halide) and reactive oxygen species (hydrogen peroxide) [10,11]. This ability to transport different solutes makes it possible to clearly identify two AQP subfamilies, the ordinary or orthodox aquaporins (AQP, *stricto sensu*), which exclusively transport water, and the aquaglyceroporins (AGP or AQGP), which are additionally permeable for these small organic and inorganic compounds [12]. Aquaporins are a ubiquitous family of intrinsic membrane proteins present throughout all kingdoms of life, from archae to eukaryotes (plants, animals, fungi, insects and protists). The plant kingdom exhibits remarkable MIP diversity, which can be classified into eight subfamilies: the plasma membrane intrinsic proteins (PIPs), tonoplast intrinsic proteins (TIPs), nodulin 26-like intrinsic protein (NIPs), small basic intrinsic proteins (SIPs), X-intrinsic proteins (XIPs), GlpF-like intrinsic proteins (GIPs), hybrid intrinsic proteins (HIPs) and large intrinsic proteins (LIPs) [13,14,15]. To date, only three full-length aquaporin sequences belonging to PIP and TIP subfamilies have been described in olive trees: *Oe*PIP1;1, *Oe*PIP2;1 and *Oe*TIP1;1 [16]. As observed in other tree species, the transcript levels of these aquaporins have been reported to be differentially down-regulated in leaves, roots and twigs from olive trees (*O. europaea* L.) subjected to drought treatment. However, a comprehensive analysis of each aquaporin member and their responses in different tissues or under various abiotic and biotic stresses is lacking.

The availability of a whole genome sequence from the wild olive species *O.* var. *sylvestris* facilitates the genome-wide analysis to identify the complete set of AQPs in *Olea* (*Oeu*AQPs) and to understand the evolutionary relationship of *Olea*’s AQPs by comparing *Oeu*AQPs with their homologuous *Oleur*AQPs from the domesticated cultivar, *O. europaea* cv. Picual. Of Andalusian origin, cv. Picual is the leading olive cultivar planted in Spain, where it accounts for 50% of national production and 20% of world production. Extra virgin olive oil from cv. Picual offers exceptional organoleptic properties and high oxidative stability due to its elevated polyphenolic compounds content. Spurred by numerous health benefits, the world demand for such a precious vegetable oil is expected to increase in the forthcoming years.

*Olea europaea* cv. Picual displays a great capacity to adapt to a wide variety of soils and climatic conditions. It allows this cultivar to be cultivated in arid plains as well as in mountain areas and to withstand very cold winters. However, like most domesticated cultivar, cv. Picual is susceptible to various plant pathogenic agents, such as the Ascomycota *Verticillium dahliae*, the causal agent of Verticillium wilt [17]. The involvement of AQPs in cellular responses to tolerate abiotic (cold and wound) and biotic (*V. dalhiae* infestation) stresses needs to be resolved. In this respect, gene expression is a major step towards the gene elucidation of its biological function, and combined with the motif patterns and related predictive substrate permeability, the plant localization of aquaporin and expression profiles provide important clues to reveal their potential physiological roles in a plant.

In this work, we performed a genome-wide analysis of *Olea*AQP gene family members in wild (*Oeu*AQPs) and cultivated (*Oleur*AQPs) olive trees, and we characterized their phylogeny, chromosomal distribution, gene structure, protein motifs and potential substrate specificities. We investigated the transcriptional expression profile of *OleurAQP* genes using RNA-seq in various organs and tissues, during the early stages of olive-seedling development and the emergence of the juvenile plant and in response to biotic (*Verticillium dahliae* root inoculation) and abiotic (root wounding and leaf chilling) stresses. The present research will contribute to bringing novel and relevant information with regards to AQP evolution and their functional regulation in olive trees during various physiological processes of development and stress responses. Our exhaustive molecular understanding on *Olea* AQPs is expected to be helpful in elucidating the importance of the MIP superfamily in a commercially eminent oleaginous crop. 

## 2. Results and Discussion

Climate changes have been highlighted as one of the most harmful events affecting the crop growth and productivity. Simultaneously, the present environmental conditions and agricultural practices create a fertile breeding ground for the spreading of a plethora phytopathogen agents. All these concomitant abiotic and biotic stresses result in significant agricultural losses worldwide. However, whatever the challenging stressors, plants must constantly adjust their water-balance control for normal development and maintenance of proper yield.

*Olea europaea* L. is part of the Lamiales, an Asterid order of core eudicots. To date, a complete picture of the AQP family from the Lamiales taxa is still missing. In this regard, the available draft genome sequence of olive trees provided us with an opportunity to analyze the AQP gene family in an oleaginous species and to assess their diversification in the *Olea* genus, and *vis-a-vis* its related Oleaceae common ash (*F. excelsior*) and some least related taxa including tomato, poplar and *Arabidopsis*.

### 2.1. Identification of the Complete Set of O. Europaea L. and F. Excelsior Aquaporins

The “Build your query” function including “Major Intrinsic Protein” and “Aquaporin” in Phytozome (V12) were used to identify the complete non-redundant AQP family of the wild olive *O. europaea* var. *sylvestris*. To investigate whether potential *Oeu*AQP genes were lacking or insufficiently annotated in the databases, the genomes of the cultivar olive *O. europaea* L. cv. Picual (OliveTreeDB website) and the Oleaceae related species *F. excelsior* (hardwoodgenomics website) were BLAST-searched using several AQP members belonging to each subfamily from *A. thaliana*, *L. esculentum* and *P. trichocarpa* as queries. Identified AQP sequences from each Oleaceae species were thereafter iteratively used as queries. Subsequently, every output putative AQP gene was manually inspected. This approach demonstrated that for the database-annotated AQP genes, output coding sequences and exon/intron selections sometimes resulted in a panel of non-satisfactory AQP gene models. With the exception of *Oeu*PIP2;11 and *Oeu*NIP5;2 sequences for which a manual annotation procedure generated very long AQP sequences, thirty-six non-satisfactory AQP gene models encode partial sequences due to an incomplete or miss-annotation procedure alongside the chromosomes or the scaffolds. These situations are linked to deletions or mutations that result in a shift in the reading frame leading to the synthesis of premature stop codons or non-functional proteins (e.g., *Oeu*NIP4;2). Wherever possible, each erroneous AQP sequence was manually corrected using orthologs from *F. excelsior* and *L. esculentum,* with the addition of AQPs from *P. trichocarpa* for the examination of the XIP subfamily. This last approach helped identify twenty new candidates encoding complete AQP sequences meeting the typical AQP features. Similarly, the alignment of the incomplete putative *Oeu*AQP sequences due to their scaffold edge location with orthologous counterparts gave three complete *Oeu*AQP sequences (*Oeu*PIP2;9, *Oeu*PIP2;10 and *Oeu*TIP4;1). Two *OeuAQP* genes were not detected and annotated in Phytozome databases (*OeuSIP2;2* on chromosome 11 and the incomplete sequence *OeuNIP4;2-related* on chromosome 8). Lastly, two raw *OeuPIP2-*related gene sequences initially emerged as two distinct 3′ and 5′ truncated AQP candidates (Oeu017254.1 and Oeu017256.1, respectively). By homology with orthologous PIP2s, these two sequences were gathered in one sequence to generate the putative *Oeu*PIP2;11 in its full-length version. However, the last intron (19421nt of size) enclosed the *Oeu*017255 gene sequence (uncharacterized protein). This sequence was retained for computational analysis since *Oleur*PIP2;11 from the domesticated olive species shares similar features.

In summary, 68 putative AQP sequences were identified in the wild variety, *O.* var. *sylvestris*. Thereof, the manual inspections and curating procedures showed that 52 sequences encode for typical full-length AQP proteins, namely, thereafter, *Oeu*AQPs. Sixteen AQP proteins had to be discarded for various reasons, and every identified *Oeu*AQP sequence is detailed in its ‘original’ version and its eventual corrected version in Appendix A and Appendix A.

In the domesticated olive cultivar ‘Picual’, 127 *Oleur*AQP protein sequences were identified. However, this high abundance needs to be put into perspective with 79 full-length and putative full-length candidates and 51 incomplete genes or pseudogenes sequences. Among the 79 full-length candidates, 26 sequences are in single copy and 26 AQP-types are duplicated, the latter generating 27 full-length sequences and 3 pseudogenes. As a result, 53 full-length AQP-types are identified in the cultivated genome, a similar AQP level was recorded in the wild variety. The *Oleur*AQP superfamily expansion and diversity will be further discussed in a specific chapter. *Oleur*AQP sequences and their correspondence with the *Oeu*AQP sequences are detailed in Appendix A and Appendix A, respectively.

Concerning *F. excelsior*, 56 putative *Fraex*AQP sequences were extracted, including 54 full-length sequences used for comparative analyses of *Olea* AQP. Sequences were extracted and verified as for *Olea* AQP sequences and specifically annotated for this work (Appendix A).

These numbers are comparable to those of species exhibiting the largest number of members. In comparison, the basal viridiplantae *Selaginella moellendorffii* and *Physcomitrella patens* show 19 and 23 AQPs, respectively. The monocotyledons *Hordeum vulgare*, *Oryza sativa* and *Zea mays* show 40, 34 and 43 AQPs, respectively. In the Rosids, the Malvids *A. thaliana* and *Brassica rapa* show 35 and 47 AQPs, and the Fabids *P. trichocarpa*, *Glycine max* and *Gossypium hirsutum* show 58, 66 and 71 members, respectively. Lastly, among the Lamiids, the two Solanales *S. tuberosum* and *S. lycopersicum* show 41 and 47 members.

### 2.2. Diversity in OeuAQP Gene Families from the Wild Olive Tree Variety, O. Europaea Var. Sylvestris

A total of 52 full-length *Oeu*AQP encoding genes were identified in the genome of the wild olive. In accordance with most of the high plant genomes, the *Oeu*AQPs were classified in five distinct subfamilies: *Oeu*PIPs (19 members), *Oeu*TIPs (17 members), *Oeu*NIPs (9 members), *Oeu*XIPs (3 members) and *Oeu*SIPs (4 members) (Figure 1 and Appendix A). Furthermore, these subfamilies can be sub-grouped based on the analysis of various motifs (i.e., the aromatic/arginine selective filters and the Froger’s residues, as detailed below) and the phylogenetic positioning that well corroborate each other (Figure 2a). Finally, the further classification of *Oeu*AQP subfamilies into subgroups shows that the var. *sylvestris* (as to the domesticated cultivar Picual or the related Oleaceae *F. excelsior*) does not contain additional phylogenetic AQP subgroups within the five subfamilies formed due to the divergence of the Asterids taxa within the core eudicots. Accordingly, *Oeu*AQP classification is consistent with AQPs from *Arabidopsis* or tomato, implying that AQP proteins have been highly conserved during evolution. In detail, the larger *Oeu*PIPs subfamily is divided in two distinct subgroups, *Oeu*PIP1 and *Oeu*PIP2, with eight and eleven members in each group, respectively.

Similarly, the *Oeu*TIP subfamily clusters in five subgroups (*Oeu*TIP1 to 5), where the *Oeu*TIP1 subgroup encompasses the greatest number of members. The subfamily *Oeu*NIPs consists of five subgroups, where the *Oeu*NIP1 subgroup encompasses the greatest number of members. No NIP2 subgroup was identified in the cultivated and wild olive trees. However, it should be noticed that the classification of this subgroup is relatively ambiguous, as observed with *Sl*NIP2;1 or *At*NIP2;1 that coalesce with various NIP4 or NIP1 members, respectively (Figure 1 and Appendix A). Using the recommendations proposed by Zou and collaborators [18], the NIP2 subgroup could specifically feature the ar/R selectivity filter (G/S/G/R), as can be observed in *S. lycopersicon* (*Sl*NIP2;1) [19], *P. trichocarpa* (*Pt*NIP2;1) [20], *Ricinus communis* (*Rc*NIP2;1) [21], *Cicer arietinum* (*Gm*NIP2;1) [22], *Jatropha curcas* (*Jc*NIP2;1) [18] and *Arachis duranensis* (*Adu*NIP2;1) [23]. This motif is distanced by 108 amino acids to the NPA domains, playing a crucial role in Si absorption in various plants species [24]. However, interesting enough, the *F. excelsior* genome has a NIP2 candidate, *Fraex*NIP2;1. This suggests that a functional divergence occurred in the Oleaceae NIP subfamily, notably by a loss of the NIP2 subgroup during the evolution of the *Olea* clade.

The *Oeu*SIP subfamily clusters the four *Oeu*SIP homologues in two subgroups, *Oeu*SIP1 and *Oeu*SIP2, with two members each. (Figure 1, Appendix A). This modest number of *Oeu*SIP members is comparable to the SIP number identified in other plant species. However, the *Oeu*XIP1 subgroup features a pseudogenization of its SIP1;3-type, unlike the domesticated cv. Picual (*Oleur*SIP1;3) or *F. excelsior* (*Fraex*XIP1;3).

Concerning the *Oeu*XIP members, they are composed of three sequences with no specific partitioning. The presence of XIPs in olive trees is particularly interesting. This XIP subfamily represents the last clade of AQPs that was discovered, mostly in plants and fungi and more marginally in the Amoeboza *Dyctyostelium discoideum*. Although this subfamily is already present in basal plant lineage such as the moss *Physcomitrella patens*, it was lost in several recent plant lineages after the divergence of the basal dicots and core dicots [25,26], where all monocots, coniferophytes and several dicots taxa (such as *Brassicaceae*) are characterized by its complete absence [27,28]. In olive trees, the phylogenetic positioning of *Olea*XIPs is in line with the diversity observed in Asterids (tomato, potato), which clearly points to that of Rosids, which exhibits at least two or three distinct subgroups [18,19,29]. In this respect, the phylogenetic organization of this XIP subclass clearly differs to that of the other AQP subclasses, clustering the XIP members in distinct taxon-specific clades, i.e., Rosids vs. Asterids, and for this last clade, Lamiales vs. Solanales. Although our comparative analysis was carried out on a small panel of XIP sequences, our results demonstrate that the evolutionary pattern of this XIP subfamily is completely unique in comparison to the other AQP subfamilies [26,27,28,29,30].

Among the *Oeu*AQPs and per subfamily, *Oeu*NIP was the most diverse subfamily with 38% sequence similarity between members at the amino acid level, whereas *Oeu*PIP1 and OeuPIP2s were the most conserved subfamilies with 89% and 79% sequence similarities, respectively (Appendix A). As for *Oeu*XIP, *Oeu*TIP and *Oeu*SIP subfamilies, they exhibit 78%, 55% and 45% sequence similarities between members, respectively. Overall, *Oeu*AQP subfamilies show an average of 29% sequences similarity between them. These values are commonly observed in most of the plant MIP superfamilies.

### 2.3. Diversity in OleurAQP Gene Families from the Domesticated Olive Tree cv. Picual’

*Oleur*AQPs are particularly abundant, with a total of 127 putative *AQP* gene sequences, including 79 full-length (Appendix A) that group into the five AQP subfamilies of PIP, TIP, XIP, NIP and SIP. Furthermore, the number of *Oleur*AQP-types in each subfamily is relatively similar to that of the wild variety (Figure 1 and Appendix A). Only two AQP-types supplement those observed in the wild variety: *Oleur*PIP2;12 (a PIP2;8/9-types analogue, also present in *F. excelsior* genome, but a priori lost in the wild variety) and *Oleur*NIP1;4.

The *Oleur*AQP superfamily expansion is linked to the fact that the Olea genome was subjected to two rounds of tetraploidy (62 and 25 million years ago) and the cultivated olive to a nearly present partial duplication event [31,32]. Ultimately, no AQP-type loss after duplication appears to have occurred. As a putative consequence to these genome rearrangements, the number of full-length AQPs in both olive varieties (79 *Oleur*AQPs for 52 *Oeu*AQPs) was well correlated with the genome size and the total number of genes, i.e., (cv. Picual vs. var. *sylvestris*) genome sizes of 1.63/1.81Gbp vs. 1.14 Gbp, encompassing 78079 vs. 50684 protein-coding transcripts. Similar abundances were also reported in other cultivated plant species that evolved with polyploidization, e.g., *Brassica napus* with 120 AQPs or *G. hirsutum* with 113 AQPs [33,34].

Gene duplication concerned all *Oleur*AQP subfamilies, in which 21 full-length *Oleur*AQP-types are represented by two full-length paralogues, except for *Oleur*TIP1;3 with 5 duplicates and *Oleur*TIP3;1 with 3 duplicates (Appendix A; Appendix A), generating 27 full-length duplicated genes sequences. On the other hand, 41 *Oleur*AQP incomplete sequences or pseudogenes add to them. Among them, six *Oleur*AQPs are related to *Oeu*AQP incomplete sequences and four pseudogene *Oleur*AQP sequences by introduction of deleterious mutations, probably during the tetraploidization process of the cultivated olive, derived from full-length *Oeu*AQP sequences.

### 2.4. Structural Features of OeuAQP and OleurAQP Proteins

Plant AQPs present different functional characteristics that are driven by specific biochemical properties [6]. The biochemical properties of the 52 *Oeu*AQPs were predicted, i.e., the molecular weight (MW), protein Isoelectric point (*p*I), grand average of hydropathy (GRAVY) and subcellular localization (Figure 3). *Oeu*AQP proteins range from 240 to 335 (average 272) amino acids in length, molecular weights range from 25.45 to 37.27 (average 28.86) kD, and *p*I values range from 5.12 to 10.04 (average 7.56) (Figure 3). Among the *Oeu*AQPs subfamilies, the lower average *p*I value concerns *Oeu*TIPs (average 5.78). This more acidic pH is mainly linked to the absence of basic residues in the *C*-terminal domains of the proteins, as also observed in *Arabidopsis* [35]. As for the GRAVY parameters, scores were all positive, ranging from 0.332 to 1.035. Positive and negative scores for protein GRAVY reflect protein hydrophobicity and hydrophilicity, respectively. As far as we are concerned here, the positive GRAVY scores of the *Oeu*AQPs indicate their hydrophobic nature, which, for an aquaporin, is a key property that facilitates high water and substrate permeability across membranes [4]. In detail, the higher GRAVY values concern the *Oeu*TIP subfamily (average 0.82). This indicates that among the *Oeu*AQP subfamilies, the *Oeu*TIPs would present a better interaction with water molecules, with the exception of *Oeu*TIP3;1 and *Oeu*TIP5;1 (0.56 and 0.64, resp.). Interestingly, the two *Oeu*SIP1 members share high GRAVY scores (0.798 and 0.83), which noticeably contrast with *Oeu*SIP2 members that show lower GRAVY values (0.417 and 0.505).

Prediction of TMHs (transmembrane helical regions) showed that the majority of identified putative *Oeu*AQPs contained six TMHs (Figure 3). However, these predictive tools did not correctly identify all TMHs, predicting six TMHs for most AQPs, but not all AQPs for which predictions are in a few cases contradicting where five or seven TMHs are predicted. That concerns all *Oeu*XIPs and *Oeu*SIPs and six *Oeu*TIP members (marked with an asterisk in Figure 3). Inspection of hydrophobicity plots (data not shown) and the amino acid sequence alignments for each AQP subfamily have therefore been necessary to remove any ambiguity (Appendix A). Given the high degree of sequence conservation between *Oeu*AQPs and different well-characterized orthologs, it is reasonable to suggest that all *Oeu*AQPs feature six TMHs (with the exception of *Oeu*TIP1;6 which presents a truncated 3′ end).

The protein motifs are highly conserved amino acid residues that are considered to possibly have key functional and/or structural roles in active proteins. Concerning aquaporins, the hydrophobicity degree and the size of the side chain of each amino acid engaging within the pores determine substrate specificity. To seek further insights into motif structures and organizations that design the *Oeu*AQPs, the 52 full-length *Oeu*AQPs proteins were analyzed using the MEME program. Fifteen conserved motifs, designated as motif 1 to motif 15, were identified (Figure 2C and Appendix A). Most *Oeu*AQPs proteins belonging to the same subfamily generally harbor similar motifs, but notably differ between subfamilies. This organization is consistent with the phylogenetic distribution (Figure 1 and Figure 2A). Similar patterns of motif conservation and/or deletion between subfamilies/subgroups have been observed in recent genome-wide studies of AQPs in other plants. Of these patterns, the *Oeu*PIP1 and *Oeu*PIP2 members present a high motif similarity but have a specific *N*- and *C*-terminal domain. These regions modulate water transport activities differentially in oocyte experiments [36,37]. Motif 4 is specifically detected in the *Oeu*PIPs subfamily, whereas motif 5 is shared between *Oeu*PIPs and *Oeu*SIP1s. Motif 3 was found in all the *Oeu*AQPs, except for *Oeu*XIPs and *Oeu*SIP2s, and motif 6 is specific to *Oeu*XIPs. In line with previous motif analysis in other plant models, the XIP and SIP subfamilies exhibit relatively few motifs. Based on analysis of conserved protein motifs between different Oleaceae and the tomato AQP, gain and loss of certain motifs can be observed within orthologous sequences/subgroups (data not shown). This highlights plausible functional divergences of some Asterid orthologues.

Some of these motifs contain specific amino acid residues (or signatures) that are crucial for the transportation function of the aquaporins (Figure 3; Appendix A), i,e., the dual Asn-Pro-Ala (NPA) (NPA1 in motif 1 and NPA2 in motif 3), the aromatic/arginine (ar/R) selectivity filter (H2 in motif 1, H5 in motif 6 and LE1 and LE2 in motif 10) and the Froger’s residues (P1 in motif 1 and P2-P3-P4-P5 in motif 7). To understand the possible physiological role and substrate specificity of olive tree aquaporins, all these residue positions were identified and analyzed.

The association of the two highly conserved “NPA” motifs initially placed in the loops LB and LE forms the main central constrict of the channel and creates an electrostatic repulsion of protons. Different point mutations of amino acids at these positions have been found to strongly impact the substrate specificities of aquaporins [38]. Every *Oeu*AQP contains these motifs in loops LB and LE, but some amino acid triplets show residue variations (Figure 3 and Appendix A). *Oeu*PIPs and *Oeu*TIP contain two NPA residue triplets. The *Oeu*XIPs members contain a conserved second NPA motif, but all of the first NPA motifs show a replacement of the alanine (A) by a valine (V). In the case of the *Oeu*SIP subfamily, members from each subgroup have dual NPA motifs, but the third NPA1 residue is replaced by a threonine (T) (*Oeu*SIP1s) or a leucine (L) (*Oeu*SIP1s). As for the *Oeu*NIP subfamily, the first NPA motif shows an alanine (A) to serine (S) substitution in three *Oeu*NIPs (*Oeu*NIP5;1, *Oeu*NIP5;2 and *Oeu*NIP6;1), and the second NPA motif shows an alanine (A) to valine (V) substitution in four *Oeu*NIPs (*Oeu*NIP1;3, *Oeu*NIP5;1, *Oeu*NIP5;2 and *Oeu*NIP6;1). Each of these substitutions partly determines the function of transporting water [39].

The aromatic/Arg (ar/R) selectivity filter is essential for selective transport of substrate molecules [2]. Compared to the two NPA motifs, the ar/R positions are characterized by high subfamily- and/or subgroup-specific residues (Figure 3 and Appendix A). The *Oeu*PIP members all contain the more hydrophilic ar/R selectivity filter (F-H-T-R), as observed in the pure water channel *Aqp*Z. This motif is a hallmark of water and solute transporting aquaporins compared to other subfamilies [5]. By contrast, these residues are highly variable in the other subfamilies, but their diversity matches each subgroup identified. In detail, among the *Oeu*TIPs, *Oeu*TIP1 members harbor residues (H/I/A/V), forming a more hydrophobic ar/R filter compared to the *Oeu*TIP2 or *Oeu*TIP3 and *Oeu*TIP4 subgroups, for which members contain an ar/R filter with (H/I/G/R) or (H/V/A/R) residues, respectively. Lastly, the three *Oeu*TIP5 members harbor the most hydrophobic (N/V/G/Y) residues. The residues present in the ar/R selectivity filter in the different *Oeu*TIP subgroups are similar to TIPs from other plant species. Among the *Oeu*NIPs, the *Oeu*NIP1, -3 and -4 subgroups which harbor the (W/V/A/R) residues would be more hydrophobic compared to the *Oeu*NIP5, -6 and -7 subgroups with (A/I/G/R), (T/I/A/R) and (A/V/G/R) residues, respectively. This suggests that each subgroup of NIPs has its own function. For example, it was reported that NIPs exhibiting a (H/V/A/R) signature present a water permeability decrease, while enhancing transport of uncharged solutes such as glycerol and formamide [40]. In addition, the NIPs that combine the ar/R signature (A/I/G/R) with the dual NPS/NPV motifs transport arsenite, boric acid and silicon in rice [41,42]. *Oeu*NIP5 members possess this combination, suggesting that the olive *Oeu*NIP5s have the ability to absorb these particular substrates. These OeuNIP5s may substitute the NIP2 subgroup that is absent in Olea genomes. *Oeu*XIPs contain a quite hydrophobic ar/R selectivity filter composed of (I-A/L/A/R). This feature is accentuated with the isoleucine and leucine at the H2 and H5 positions. The ar/R selectivity filter in XIPs from different plants is relatively hydrophobic in nature, facilitating the transport of bulky and hydrophobic molecules such as glycerol, urea and boric acid in plants [30,43,44,45].

Froger’s positions concern five residues that have an important role in the selection of molecules across cell membranes, helping to discriminate between aquaglyceroporines (AQGP) and aquaporins (AQP *stricto sensus*) [46]. These Froger’s residues concern an aromatic residue at P1, an acidic residue at P2, a basic residue at P3 and two non-aromatic residues at P4 and P5. A relative consensus can be observed between the *Oeu*AQP subfamilies, but some singularities emerge (Figure 3 and Appendix A). The *Oeu*PIP members exhibit identical amino acids in the four P2‒P5 residues (S-A-F-W), but the first residue is highly variable with three amino acids, glycine (G), methionine (M) or glutamine (Q), which are alternatively encountered at this position. As for *Oeu*TIPs, the five residues (T/S/A/Y/W) are highly conserved, with the exception of *Oeu*TIP3;1 and *Oeu*TIP5;3, for which P2 is an alanine (A). The Froger’s positions (as in the ar/R selectivity filter) are quite divergent in the *Oeu*TIP subfamily members compared with those in other subfamilies, indicating that *Oeu*TIPs may have different solute permeabilities. The three *Oeu*XIPs harbor identical residue positions, as well as members belonging to each *Oeu*SIP subgroup. Concerning the *Oeu*NIP members, a great variability appears between members but with subgroup-dependent patterns, e.g., *Oeu*NIP1s with (F-S-A-Y-I), *Oeu*NIP3 with (F-S-A-Y-V), *Oeu*NIP4 with (F-S-A-Y-I), *Oeu*NIP5s with (F-T-A-Y-L) and *Oeu*NIP6 with (Y-S-A-F-L).

The different steric hindrances of the side chains involved in the variable NPA motifs, ar/R selectivity filter and Froger’s observed between *Oeu*AQPs form differential constriction zones. Their comprehensive comparison would indicate that *Oeu*PIPs, *Oeu*TIPs, *Oeu*XIPs, *Oeu*NIPs and *Oeu*SIPs facilitate the transport of distinct substrates across membranes. Interestingly, they coordinately play a crucial role in facilitating the transport of various permeants with some other amino acid residues, for which their own side chain is involved within the central pore [6,18,21]. To this respect, the prediction of *Oeu*AQP functions, based on key protein domains conservation (SPD) as proposed by Hove and Bhave [6], was recorded (Figure 4; Appendix A). *Oeu*PIPs have H_2_O_2_-type (with the exception of *Oeu*PIP2;11) and urea-type SDPs. In addition, all *Oeu*PIP1s have boric acid-type SDPs, and only two of them, *Oeu*PIP1;4 and *Oeu*PIP1;6 display CO_2_-type SDPs. These results show the conservation of plant PIPs in the transport of urea and hydrogen peroxide [47,48] and that PIP1s—but not PIP2s—could be main boric acid channels and CO_2_ [49]. Concerning CO_2_, AQPs were shown to facilitate the diffusion of gases, and as such, AQP activity may affect CO_2_ transport and carbon metabolism as a whole [50,51]. Compared to *Oeu*PIPs, the *Oeu*TIP subfamily that features specific and conserved selectivity filters between members all have urea-type SDPs, and four members (*Oeu*TIP1;7, *Oeu*TIP2;1, *Oeu*TIP2;2 and *Oeu*TIP5;3) harbor H_2_O_2_-type SDPs. The fact that the *Oeu*TIP subgroups harbor specific ar/R filters and SDP signatures suggests that *Oeu*TIPs play a crucial role in transporting a wide range of molecules, as experimentally observed in *Citrus sinensis* [6].

The NIP subfamily has low intrinsic water permeability and the ability to transport solutes such as glycerol and ammonia [52]. All *Oeu*NIPs have urea-type SDPs, except for OeuNIP6s. In addition, *Oeu*NIP1;1 has the H_2_O_2_-type SDPs. Similarly, *Oeu*NIP5s and *Oeu*NIP6;1 members (with the presence of a glutamic acid (E) in SDP9 for which its regulatory function needs to be experimentally validated) have boric acid-type SDPs. This last transport prediction is in accordance with the NPS/NPV aqueous pore and (A-I-G-R) ar/R selectivity filters observed in these NIP members, which are known as boric acid transporters in the orthologous *At*NIP5;1 [42]. This suggests that NIPs could be involved in the transport of larger solutes. SDP analysis suggests that the *Oeu*XIP subgroup displays contrasted solute permeability between members: *Oeu*XIP1;1 is a transporter of boric acid and H_2_O_2_, *Oeu*XIP1;2 is a transporter of boric acid and urea and *Oeu*XIP1;3 is a transporter of boric acid, H_2_O_2_ and urea. These SDP signatures corroborate 3D modeling and functional analysis [43,44]. Notwithstanding that their functional role remains quite unclear in plants, it is possible that *Oeu*XIPs have different roles in olive trees.

It should be noted that we were unable to predict SDP signatures for the four *Oeu*SIP members. Similarly, no *Oeu*AQP held silicic acid-type and NH_3_-type SDPs. We have already reported that the NIP2 subgroup is absent in the olive genome. However, it has been reported that the NIP3 subgroup is also a transporter of Si (and Ge, As and B) and that their expression can be a critical factor in determining the ability of a plant to absorb Si [24,25,26,27,28,29,30,31,32,33,34,35,36,37,38,39,40,41,42,43,44,45,46,47,48,49,50,51,52,53]. *Oeu*NIP3;1 could assume this role of Si transporter. Incidentally, it suggests that other types of SDPs could be involved in Si absorption (in particular for *Oeu*NIP3;1) during plant growth, development or stress responses in olive trees. As for the NH_3_ element, no complete NH_3_-type SDPs were identified in the two concerned analogous *Oeu*TIP and *Oleur*TIP subgroups, whereas they are present in their counterpart *Fraex*TIP2;1 (T/L/T/V/S/H/P/A), as confirmed with its orthologous *At*TIP2;1 [54] (data not shown). This functional divergence shows the nucleotide substitutions that played a crucial role in the diversification of the conserved residues that determine the substrate specificity. It supports the assertion that these evolutionary impetuses are particularly universal in the AQP superfamily, reflecting a wide variety of their complementary substrate transport capacity.

Aquaporin activity is regulated transcriptionally and/or post-translationally by pH, phosphorylation and vesicle trafficking. In this context, a panel of motifs and post-translational regulation signatures were identified in *Oeu*PIPs. It is admitted that PIP1s have null or lower water permeability than PIP2s and that a functional difference would be attributed to one residue just after H2 in the THM2: all PIP1s possess alanine (A) while the PIP2s have valine (V) or isoleucine (I) [55]. At this position, *Oeu*PIP1s and *Oeu*PIP2s feature this singular difference (Appendix A), suggesting that the *Oeu*PIP2 subgroup is more permeable to water than the *Oeu*PIP1 subgroup. This hypothesis is supported by the observation of Secchi et al. [16] that *Oe*PIP2;1 (the *Oeu*PIP2;1 analogue) is highly permeable to water while *Oe*PIP1 has no water transport activity in oocytes. Several *Oeu*TIPs share this alanine residue with *Oeu*PIP1s, though they are considered (at least in theory) as being water permeable. However, previous data showed that *Oe*TIP1.1 (the *Oeu*TIP1;1 analogue) exhibits water transport activity in oocytes, though lower than *Oe*PIP2 [16]. It is evident that water permeability is defined by more than one residue. In this respect, some *Oeu*TIPs such as *Oeu*TIP1;3 and *Oeu*TIP2;1-2-3 harbor an alanine substituted with a glycine (G) (Appendix A). Interestingly, these members show the highest levels of transcriptional expression (*cf* RNAseq expression item). With the exception of *Oeu*NIP7;1, *Oeu*NIP members also exhibit this glycine position (Appendix A). The plausible contribution of this small glycine to the water permeability of *Oeu*TIP and *Oeu*NIP members has to be resolved.

Similarly, AQPs are known to function as tetramers. For this to happen, the highly conserved cysteine (C) (Cys80 in *Zm*PIP2;1) in loop LA of PIP1s and PIP2s is shown to be essential for formation of disulphide bonds between PIP monomers, increasing the oligomer stability [56]. Every *Oeu*PIP1 and *Oeu*PIP2 contains this cysteine (Appendix A), suggesting possible oligomerization between members. However, this specific residue is absent in TIPs, XIPs, NIPs and SIPs, suggesting that other residue(s) could be involved in the structural relationship of monomers leading to AQP quaternary models.

Furthermore, we note the presence of various phosphorylation sites in all *Oeu*PIP1 and *Oeu*PIP2 members with, for example, the serine (S) residue in the loop LB that is located in domains which correspond to the recognition sequences of protein kinase A (R-K-x-S) and protein kinase C (R-K-x-S-x-x-K) in all *Oeu*PIP1s and *Oeu*PIP2s, with the exception of *Oeu*PIP2;10, which contains a calmodulin-like domain protein kinase (L-x-R-x-x-S) (Appendix A). In addition, the *C*-termini of some OeuPIP2s have a serine located in the potential cAMP-dependent protein kinase site (S-x-K). Lastly, different sites can be observed, such as the *N*-t AEFxxT-box at the -60 amino acid position to the first NAP with unclear function, the histidine (H) residue in loop LD involved in gating, the *N*-terminal diacidic motif KD(V/I)E for putative conserved methylation sites and found to be important for the exit of newly synthesized PIPs from the ER to plasma membrane and the putative conserved blocking leucine residue (L) in TMH5. Regarding the NIP subfamily, different phosphorylation sites are also found in the *C*-termini of *Oeu*NIP1;1, *Oeu*NIP1;3 and *Oeu*NIP4;1 (Appendix A). It would then be very interesting to specifically mutate some of these residues in order to validate the importance of these regulation points in the permeability control of these AQPs.

In Picual cultivar, 77% of *Oleur*AQP-types have >99% similarity with their *Oeu*AQP paralogues. For the other *Oleur*AQP-types, and despite several residue variations, all *Oleur*AQP-types adhere to the *Oeu*AQP structures and features. Most *Oleur*AQPs contain residue signatures (ar/R filter, Froger’s position and the NPA and different substrate-type SDP motives) identical to those of their *Oeu*AQP paralogues. Only *Oleur*PIP2;2, *Oleur*PIP2;6, *Oleur*TIP5s, *Oleur*NIP6;1 and *Oleur*SIP1;1c diverge in some of the residues constituting the ar/R filter, Froger’s positions and NPA motives. In addition, *Oleur*PIP2;2ab and *Oleur*NIP7;1ab display residue substitutions in their urea-type SDPs, annihilating the urea transport capacity of these members, a priori.

Although some variations in amino acids are seemingly fixed between paralogues during the evolution of the Oleaceae, strong sequence similarities and protein functionalities seem to be conserved between *Oleur*AQPs and *Oeu*AQPs, suggesting that each paralogue contributes to specific and common key biological functions in a domesticated cultivar. This hypothesis is strengthened by the purifying selections that predominated across the paralogue gene pairs between wild and domesticated olive trees. This evolution pressure of domestication might have resulted in the conservation of protein sequences and related biological functionalities.

In conclusion, the overall protein similarity between AQPs from the Oleaceae species studied here and a large panel of plant species, notably at the residues constituting the different key motifs designing the permease selectivity and ultimately the biological functions, suggests that the knowledge on channel selectivity which was revealed in various plant models might be easily transferable to Oleaceae crop isoforms. This would indicate that the diversity of AQPs in *O. europaea* L. have a crucial role in the cell responses to environmental stresses.

### 2.5. Localization of the OeuAQP Proteins

Investigations have been increasingly deployed to understand tissue and subcellular specificity of genome-wide AQPs in plants, especially as the subfamily names do not systematically indicate location [reviewed in 37,57]. To this respect, sub-cellular localizations of *Oeu*AQPs were predicted to ascertain expression at different cellular and organellar levels (Figure 3).

Plant AQPs exhibit a broader subcellular localization, where PIPs, NIPs and XIPs are usually found in the plasma membrane or chloroplast membranes, while TIPs and SIPs are generally localized to the tonoplast and endoplasmic reticulum and Golgi apparatus, respectively [2,43]. The subcellular localization prediction of *Oeu*AQPs showed that, as observed in other plants and suggested by their nomenclatures, basic *Oeu*PIPs and acidic *Oeu*TIPs are predicted to preferentially localize to plasma membranes and vacuolar membranes, respectively.

In addition, predictions situate some *Oeu*PIPs at the vacuolar membrane and/or endoplasmic reticulum, and *Oeu*TIPs to plasma membranes. All basic *Oeu*NIPs were predicted to preferentially target the plasma membrane, whereas *Oeu*TIP1;6–7 and *Oeu*NIP4;1 appeared localized in the tonoplast. Their NIP homologs in several organisms were determined to localize to plasma membrane, endoplasmic reticulum or the peribacteroid membrane of root nodules [42,43,44,45,46,47,48,49,50,51,52,53,54,55,56,57,58].

Compared with the diverse predicted localizations using WoLF PSORT or Plant-mPLoc, all *Oeu*XIPs were predicted to localize predominantly to the plasma membrane, which is consistent with other experimental results [43,44].

Concerning the last *Oeu*SIP subfamily we studied, a wide range of subcellular localizations are reported, including plasma membrane, tonoplast, endoplasmic reticulum, Golgi apparatus, nuclear membrane and chloroplasts. It has also been reported that SIP1 proteins could transport water across endoplasmic reticulum membrane while SIP2 proteins may act as an endoplasmic reticulum channel for small molecules or ions [59,60,61].

Overall, many experimental observations report that AQP subcellular localizations are multiple, complex and finely regulated. For example, *Nt*AQP1, a member of the PIP group from tobacco, was found in the inner chloroplast membrane and the plasma membrane [51]. Dual localizations have also been observed for *Zm*PIP1;2 from maize in the plasma membrane and endoplasmic reticulum [36] and the seed specific *At*TIP3;1 and *At*TIP3;2 isoforms from *Arabidopsis*, which were observed in tonoplast and plasma membrane [62]. It is now clearly established that the subcellular localization, relocalization and redistribution of specific AQPs are influenced by fluctuating environmental conditions [63,64]. However, the molecular mechanisms underlying the multiple localizations of these genes are not yet clear.

Our in silico analyses are highly informative but not conclusive. Although our predicted subcellular localizations of most *Oeu*AQPs are consistent with the general features of plant AQPs, further investigations are required to better document the subcellular localizations of *Oeu*AQPs.

### 2.6. Chromosomal Distribution and Gene Structure of OeuAQPs

The Olea genome is constituted of 23 chromosomes. The physical mapping of the 68 identified *OeuAQP*-encoding genes (full-length and pseudogenes) indicates that they are distributed on nineteen chromosomes (Figure 5). With the exception of chromosome 4, which contains the largest number of *OeuAQP* genes with seven members, other chromosomes harbor one to three *OeuAQP* members. No *OeuAQPs* gene was found on chromosomes 7, 14, 15 and 23. To date, 22 *OeuAQPs* are positioned on 22 scaffolds not yet assigned to a chromosome, and that concerns particularly the *Oeu*PIP2 subfamily with seven members out of the eleven *OeuPIP2s* identified. *OeuAQP* gene density varies on individual chromosomes. However, globally, *OeuAQP* genes were dispersed in a single manner alongside the chromosomes. However, we note five clusters on the chromosomes 2, 4, 12, 19 and 20, which correspond to tandem duplication of *OeuTIP2;1*/*OeuTIP2;2* (chr2), *OeuTIP1;4-OeuTIP1;5* (chr4); *OeuTIP5;1-OeuTIP5;2* (chr12), *OeuPIP1;1a-b-c* (chr19) and *OeuTIP1;2-OeuTIP1-related* (chr20). In addition, we found two *OeuNIP3s* (*OeuNIP3;1-OeuNIP3-related*) tandemly duplicated in the unanchored scaffold 635. All these paralogues are characterized by the same open reading frame orientation on the same linkage group, and duplicates are relatively close to each other (*Oeu*PIP1a-b-c: 2897/7516nt, resp; *Oeu*TIP1;4-5: 1702nt; *Oeu*TIP2;1-2, 1159nt; and *Oeu*TIP5;1-2, 1022nt).

Compared to the 68 cv. Picual *Oleur*AQP duplicates, the number of *OeuAQP* gene pairs is relatively small. Nonetheless, gene duplication is considered as a major driving force for the evolution and the expansion of gene families, which may also contribute to stress tolerance via gene dosage increasing, avoiding some deleterious mutations and creating the opportunity for new function emergence [65]. In this regard, several duplication events have been recorded during the evolution of several plant species leading in particular to the expansion of the MIP superfamily [66,67]. The tandemly duplicated genes mostly concern the *Oeu*TIP subfamily, and particularly the *Oeu*TIP1 subgroup. That suggests that tandem duplications might have played a vital role in the expansion of the AQP gene family in olive trees, affecting in particular the TIP subfamily.

Furthermore, the duplicated gene pairs might have undergone three alternative fates during their evolution, i.e., nonfunctionalization, neofunctionalization and subfunctionalization [68]. Our results show that each full-length *Oeu*AQP duplicate has a *K*a/*K*s ratio between 0,001 and 0,2201, suggesting that *Oeu*AQP genes in the wild variety might have experienced strong purifying selection pressure with limited functional divergence after duplication. Purifying selection supports the process of elimination of deleterious loss-of-function mutations, enhances fixation and preserves the function of a new duplicated gene [69]. Our results suggest that the duplicated *OeuTIP* genes in the wild olive species did not diverge significantly during subsequent evolution, where purifying selection might have contributed greatly to the maintenance of function in the TIP subfamily genes. However, the exact functional contribution for each paralogue needs to be established.

By contrast, it should be noted that the three *OeuXIPs* are dispersed on the chromosomes 3, 5 and 16. Although the chromosome-scale assembly is not yet available for the olive cultivar Picual or for *F. excelsior*, the unanchored scaffolds of these species are wide enough on both sides of the *XIP* sequences to confirm the absence of tandem duplicates (data not shown). This distribution contrasts radically with what is generally observed in plant genomes which contain this singular subfamily and where the *XIP* paralogues are systematically found next to each other on chromosomes (including tomato [19]). Would this mean that XIP genes differentially evolved in Lamiales? Only access to additional related genomes will answer this intriguing question.

The exon-intron structure organization is an additional lever to provide clues for understanding the evolutionary connection of a gene family. In the three Oleaceae families, the number of AQP exons ranged from two to five. Although the length of each exon is similar for most members in each subfamily, some deviations are noted. In *O. europaea* var. *sylvestris*, with a few exceptions, the number and the size of the exons are conserved within each AQP subfamily (Figure 2B). This finding further validates the nomenclature we proposed with the phylogenetic analysis (Figure 1 and Figure 2A). However, these features differ notably between AQP subfamilies. Most *OeuPIP* members are characterized by four exons, the exceptions being *OeuPIP2;3* and *OeuPIP2;9*, which feature only three exons. The genes assigned to the *Oeu*TIP subfamily are mostly composed of three exons. Only *OeuTIP1;3* contains two exons. As for *Oeu*TIP4;1 and its analogous *Oleur*TIP4;1, the second intron is remarkably long, with 6483nt and 6041nt, respectively. This particularity is not shared by *Fraex*TIP4;1 (315n) and seems specific to *Olea* genomes. In addition, the majority of the *OeuNIP* members contain five exons, except for the two *OeuNIP5s*, which harbor four exons due to the lack of the second intron. Another observation can be made with this NIP5 subgroup: while in *Oeu*NIP5;2 the exon and intron lengths are comparable to different orthologous NIP5, *Oeu*NIP5;1 isoforms possess longer introns, giving a *OeuNIP5;1* gene size of more than 14032 bp (the same is observed with *OleurNIP5;1a*, 7392nt; *FraexNIP5;1*, 5999nt). Despite these intron elongations, the sequence identity of *Oeu*NIP5;1 exons is similar to other NIP subgroups, resulting in highly similar NIP5 protein sequences. As observed with the TIP4 members, the molecular and evolutionary reasons for intron elongation tendencies have yet to be resolved. The subfamily of *Oeu*SIPs are characterized by a conserved three-exon structure. However, this apparent conservation is relatively unexpected, particularly with respect to the very low identity percentage between the two *Oeu*SIP1 subgroups. As for the last XIP subfamily, the *OeuXIP* members display three exons (*OeuXIP1;1*) or four exons (*OeuXIP1; -2* and *-3*). The exon structure of the XIP subfamily is relatively variable between plant species.

By contrast, the introns vary greatly in both length and position. This is particularly marked for the *Oeu*XIP, *Oeu*NIP and *Oeu*SIP subfamilies, and with the *Oeu*PIP exception of *OeuPIP2;11*, which is split into two parts by the integration of a putative gene. That results in a gene length within *Oeu*AQP that varies from 0.86Kb (*Oeu*TIP1;3) to 14.032Kb (*Oeu*NIP5;1), corresponding to the exon-intron patterns observed in other crop or fundamental plants belonging to distinctly different taxa such as *Arabidopsis* [35], soybean [70], orange [29], poplar [20], chickpea [22] or watermelon [71].

### 2.7. Gene Structure of OleurAQPs

The ‘Picual’ cultivar genome encompasses 79 full-length sequences, including 26 single sequences and 53 duplicated gene sequences (Appendix A). Concerning the duplicates, only four sequences show 100% similarity with their paralogue (Appendix A). However, although other paralogues show several variations in nucleotides, all protein duplicates share identical residue signatures and motifs to their paralogues. This is consistent with a nearly present duplication of this cultivar. Thus, it has taken a very short time to diverge, suggesting that they retained the same biological function in the course of evolution. Only two paralogues form an exception, *Oleur*PIP1;3b and *Oleur*NIP7;1b, which display substituted residues at the P5 and P1 Forger’s positions, respectively.

Similarly, the intron-exon organizations are highly conserved, although there are some differences in the intron size between the wild and domesticated species or between several *Oleur*AQP duplicates. In addition, some particular exon divisions for *Oleur*PIP1;3a, *Oleur*PIP2;9, *Oleur*TIP1;7, *Oleur*XIP1;1b and *Oleur*NIP6;1b are observable, without impacting the final protein structures. The exon-intron structural divergences commonly occurred in the evolution of duplicate and paralogous gene sequences, and notably, the number and the size of the introns are correlated with the gene expression, gene duplication and gene diversification in plants [28]. These intragenic reorganizations occur through possible mechanisms of gain/loss, exonization/pseudo-exonization and insertion/deletion events [72]. Nevertheless, this intron diversity in the Oleaceae clade suggests original avenues of investigation.

Overall, gene duplication is a major mechanism for acquiring new genes and creating genetic novelty in an organism, enabling plants to rapidly adapt to new environments [73]. As with *Oleur*NIP1;4, each *Oleur*AQP duplicate is to be considered as a new sequence. However, the analysis of phylogenetic relationships between functional domains, exon/intron structure and gene duplications showed the relatively high conserved evolution of olive AQPs, and in this respect, the average *K*a/*K*s value between *Oleur*AQP duplicate pairs (i.e., 0.1637) is very low (except for *Oleur*NIP7;1b, which is under a diversifying selection with a ratio of 1.1560) or even smaller than the evolutionary dynamics between *Oeu*AQP and *Oleur*AQP homologues (i.e., 0.4183) (except for AQP2;10, TIP1;6; TIP1;7, XIP1;3 and NIP7;1b, which are under diversifying selection).

All these results suggest a relatively lower rate of evolution of the AQP-type (including the duplicate members) in both *Olea* species, which has resulted in a very recent duplication with very little time to evolve, preserving their potential biological functions, a priori.

### 2.8. Transcriptional Profiling of Olive OleurAQPs

*Olea* in general and in particular cv. ‘Picual’ adapt well to a wide variety of soils and arid environmental conditions. Timely up- or down-regulation of certain aquaporin isoforms in normal physiological processes and under stress conditions is thought to be important for acclimation and/or tolerance to stresses. For this purpose, *Oleur*AQP transcriptional levels were monitored using RNA-seq in various organs (fruits, flowers, leaves, roots and stems) and tissue (meristems). In addition, the *Oleur*AQP expression kinetics were recorded from the early stage of growing young seedlings to the emergence of the juvenile plant and under three stress conditions, wound and *V. dahliae* inoculation on roots and cold stress on leaves (Figure 6).

As previously discussed, 21 *Oleur*AQP-types are represented by two duplicates or more. But whatever the variability level between counterparts, RNaseq analyses reveal a transcriptional expression for each, except for *Oleur*TIP1;3c, *Oleur*TIP7;1b, *Oleur*XIP1;1b and *Oleur*NIP5;1b, which show null expression. Every *Oleur*AQP-type with its full-length duplicate sequence is presented here. However, we shall develop our arguments mainly on the expression data from the *Oleur*AQP-type, and this for two reasons: (i) even if the paralogues for a *Oleur*AQP-type show significant variabilities in their transcriptional expression, the kinetic of expression (i.e., up or down profiles) are quite similar; (ii) we are analyzing the variations in *Oleur*AQP expression only with respect to the potential functional roles the related proteins would play in different physiological processes in olive trees. Concerning the pseudogenes and incomplete sequence, all show null expression in any condition (Appendix A). This confirms the pseudogenization of these sequences (if misassembled, they might show some expression patterns). Thus, they were taken out of the discussion.

The olive *AQP* gene family displays complex differential expression patterns according to organs or tissues. Among the six organs tested, fruits, flowers and roots showed the highest levels of AQP abundance, thereby indicating their potential physiological function in these tissues during intense developmental or mineral and/or water absorption processes. In this regard, thirty *OleurAQP*-type genes significantly express in at least one organ. However, some exclusive or highly confined expressions of some of the AQPs in specific organs can be observed. In addition, every *OleurAQP*-type gene exhibits differential modulation during plantlet development or biotic and abiotic stresses.

The two subfamilies *Oleur*PIP and *Oleur*TIP show the highest expression in all organs compared to other subfamilies. *OleurPIP1s* ubiquitously express but with strong variation between members, whereas *OleurPIP1;1* is the most expressed member in all organs (RPKM ≈ 403), except in roots where *OleurPIP1:3* shows the highest expression (RPKM ≈ 266). *OleurPIP1;6* shows no expression, except for the first month of plantlet development where it expresses slightly (RPKM = 31). Five *Oleur*PIP2s, *OleurPIP2;1-2-5-8* and *-12*, present the strongest transcript accumulations in all organs (RPKM ≈ 75), but with strong differential expression levels according to the organs. *OleurPIP;12* is the most common PIP2 member expressed and it accumulates in all bioassays (RPKM ≈ 155). In terms of genome evolution, the case of this isoform is interesting because the loss of *Oleur*PIP;12 in the wild variety suggests that its functions were shared with other AQPs. *OleurPIP2;6* is expressed in most organs (RPKM ≈ 70), except in roots where it expresses slightly (RPKM = 14), while *OleurPIP2;8* mainly expresses in flowers and roots. *OleurPIP2;1* seems to be mainly expressed in roots.

These contrasted transcript accumulations can reveal very interesting plausible organ-specific compensations between protein isoforms, for example *Oleur*PIP1;3 with *Oleur*PIP1;6 and *Oleur*PIP2;6 with *Oleur*PIP2;8. In addition, members of PIP1 and PIP2 subgroups can associate together in heterodimers and tetramers. This association enhances water transport, in particular for PIP1 members which originally had no water permeability [74,75]. PIP heteromerization events can be then viewed throughout the expressions of some *OleurPIP* counterparts that are well-correlated, i.e., *OleurPIP1;3* with *OleurPIP2;8* and *OleurPIP1;6* with *OleurPIP2;6*. The fact that these *Oleur*PIPs (as their paralogous *Oeu*PIPs; Appendix A) display the conserved cysteine residue in loop A, which is essential for the formation of the disulphide bonds between PIP monomers, could hypothetically underpin, in part, the reason of such compensatory organ-dependent co-expressions observed for a multigenic family. Three-dimension simulating prediction coupled to *P_f_* calculation and oocyte experiments would bring relevant biochemical information to these protein heteromerizations.

During early plantlet development stages or stress conditions on roots and leaves, *OleurPIP1;1* is significantly up-regulated in any biological conditions, while *OleurPIP1;2*, *OleurPIP1;4*, *OleurPIP2;1* and *OleurPIP2;12* are strongly repressed. *OleurPIP1;3* is up-regulated during plantlet development and wounding, up-regulated at 48hpi and then repressed during *V. dahliae* inoculation and by cold stress. *OleurPIP2;5* is up-regulated during plantlet development and *V. dahlia* inoculation but repressed by wound and cold stress. *OleurPIP2;8* is up-regulated by wounding and by *V. dahliae* and repressed at four months of plantlet development and cold. *OleurPIP2;6* seems to be only up-regulated by cold. These contrasted profiles demonstrate, once again, the complex and plural fine-tuned regulations of AQP in plants.

In the *Oleur*TIP subfamily, only nine *OleurTIP*-types are expressed in all organs but in very imbalanced levels. *OleurTIP1;2* and *OleurTIP1;3* are the highly expressed members (RPKM > 500), while *OleurTIP1;1* and *OleurTIP4;1* present infinitesimal expression (RPKM < 10). However, the confined expressions of some of the TIP subfamily in specific organs are more pronounced than for the PIP subfamily. *OleurTIP1;3* expresses mainly in fruits, roots and stems, while *OleurTIP2;1* and *OleurTIP2;3* are expressed in fruits, flowers and roots. *OleurTIP1;6* expresses in fruits, flowers and leaves, and *OleurTIP4;1* mainly expresses in fruits. During plantlet development or stress conditions on roots and leaves, *OleurTIP1;3* expression is maximal in the first month of the platelet emergence (RPKM = 1318), and then decreases progressively (RPKM = 137). Concerning the stress conditions for the most significant, *OleurTIP1;3a* expression is repressed drastically (RPKM_essai_ = 162/114 vs. RPKM_control_ = 64,5), and ultimately, it is up-regulated significantly (RPKM=751), while the expression of *OleurTIPde* duplicates are up-regulated (RPKM_essai_ > 150 vs. RPKM_control_ < 100), except in leaves subjected to cold stress where they are all up-regulated. *OleurTIP1;2* shows similar kinetics in wounded roots and roots inoculated with *V. dahliae*. As for the *OleurTIP1;2b* duplicate, it highly expresses in fruits and moderately in roots and leaves. During abiotic stress in roots, this duplicate is up-regulated, while it is repressed by cold. Previous studies have shown a substantial role for TIP3 in flowering and seed development, specifically in the desiccation process [21,33,76,77,78]. As for the three last *Oleur*AQP subfamilies, *OleurNIP*, *OleurXIP* and *OleurSIP*, they show relatively very low expression levels (RPKM < 50 in most biological situations). In the *Oleur*NIP subfamily, the NIP transcript levels are lower than other MIPs, which is consistent with those observed in several plant species. *Oleur*NIP7;1 is strongly expressed in flowers (RPKM > 378), and *Oleur*NIP5;1 and *Oleur*NIP5;3 are expressed in roots and meristems (RPKM = 40 and 35, resp.). Interestingly, *At*NIP5;1 expression and function are linked to boric acid uptake into the roots of Arabidopsis [42]. This suggests that *Oleur*NIP5 channels play a potential role in boron transport in organs other than the root of olive trees. Concretely, although the water transport capacity of NIPs is rather limited, several NIP members are reported to be crucial for plant growth and development under boron limitation or in response to various abiotic stresses. In this regard, their biological roles in olive trees are worthy of further confirmation. Lastly, *OleurXIP1;1* is mainly expressed in flowers, roots and leaves (RPKM ≈ 35), and *OleurSIP1;1* is ubiquitously expressed (RPKM ≈ 35), with a preferential expression in flowers (RPKM ≈ 100).

In our analysis, *Oleur*TIP3s genes, which are represented by three duplicates, showed null expression in all tissues. That includes flowers, but even more during the initial phases of seed germination, for which the TIP3 subfamily is showed to be specifically and/or strongly expressed [62,76,77,78]. These results suggested that TIP3’s gene function is not strictly conserved in plants, and this assertion was also reported in *Jatropha curcas* L. [18], banana [79] and carnation [80]. It is reasonable to hypothesize that other AQP proteins may replace this TIP3 function in flowers in several plant species, including olive trees. *OleurTIP1;3* or *OleurTIP2;3*, which showed high expression in flowers, could be the alternative counterparts. These results underpin the scheme that the functions of the same subgroup are differentiated in different species and that the aquaporin family is functionally redundant.

### 2.9. Functional Consideration of OleurAQP

The ubiquitous and high expression of PIP and TIP in various organs and tissues is widely reported [18,81,82]. In this regard and whatever the stress, reduced expression of these members is generally expected to prevent loss of metabolic energy in situations of severe stress and/or to prevent loss of water from challenged organs or tissues to the hypertonic surrounding environment. However, a global analysis of these subfamilies, as for *Oleur*PIPs and *Oleur*TIPs, shows concomitant up- and down-regulations between members. These differential and spatio-temporal gene responses indicate that AQP might perform diverse functions in a variety of biological situations and that AQPs operate together as a network, probably due to the multigenic diversity of members and possible compensation mechanisms between close homologs [83,84]. For example, during *V. dahliae* root inoculation, most AQP expressions are significantly down-regulated, except for *OleurPIP1;3*, *OleurPIP2;5*, *OleurPIP2;8*, *OleurTIP1;3* and some *OleurTIP2s* which are up-regulated. Plant pathogen development depends on the host for nutrition and water. Therefore, water movement at the sites of infection is crucial for disease progress, and limiting its availability could be interpreted as a potential defense mechanism. AQPs have rarely been considered as key candidates in biotic stress studies, the pathogenicity-related genes and their host counterpart resistance genes and disease response genes being the attractive targets. However, one of the efficient cell defense responses is linked to an oxidative burst with the production of reactive oxygen species (ROS). Among ROS, H_2_O_2_ is a signaling molecule involved in various cell responses, including plant disease immunity, and its transport across membranes is provided by specific aquaporines (also called peroxyporines in such a context) [85,86]. Despite ‘Picual’ cultivar being quite susceptible to *V. dahliae*, challenged roots significantly overexpressed some *Oleur*APQs (*Oleur*PIP1;3 and *Oleur*PIP2;8 at 48hpi, or *Oleur*TIP2;1 and *Oleur*TIP2;2 at 15dpi) that display H_2_O_2_-type SDPs (Figure 4 and Figure 5), and for which transcriptional expressions correlate well with proteins directly involved in ROS protection [87]. Likewise, RNA-seq analyses clearly show several overexpressed *Oleur*PIP and *Oleur*TIP during cold acclimation. This biological phenomenon is the result of several biochemical processes, such as the de novo synthesis and the cell exchanges of cryoprotectant molecules, maintenance of ion homeostasis and a stress cell response mainly mediated by the scavenging and the removing of ROS [86,88,89]. Owing to their ability to transport such permeant cryoprotectant molecules, various AQPs belonging to PIP and TIP subfamilies would function in concert to regulate membrane water and solutes permeability, a key step in tolerating cold acclimation and freeze-induced dehydration stress [90,91]. Further studies are needed to elucidate the direct role of *Oleur*AQPs in biotic and abiotic stress regulations, in particular with regard to ROS.

A similar global trend is observed during the early stages of the olive-seedling development where most *Oleur*AQP are up-regulated. Cell division and expansion require the continuous uptake of water to maintain turgor pressure. This process is controlled by a gradient in water potential, which itself is generated by the accumulation of solutes. In this regard, cell and tissue hydraulic properties during expansion need to be tightly regulated, and the significance of PIP and TIP aquaporins in tissue elongation implies a high hydraulic permeability of the plasma membrane and tonoplast. In addition, seed germination, embryonic development and the emergence of the juvenile plants are eminently intricate physiological processes [92]. The functional roles that some *Oleur*PIPs and *Oleur*TIPs could perform in these successive plant life processes in olive trees need to be exhaustively resolved.

### 2.10. AQP in the Evolution Course of the Olive Tree Domestication

The domestication process allows plant species to artificially change their phenotypes, aiming to obtain traits human beings find desirable. Several genome rearrangement processes are observed, but duplicate genes derived from polyploidy and their ultimate potential divergence during the evolution of the “novel” plant species can be the key to crop domestication and the evolution of various phenotype traits (development, maturation or stress tolerance) [93,94]. Divergence of duplicated gene pairs is a pervasive evolutionary process after duplication events, and the recent duplication of the cultivated olive genome gave rise to a panel of very close AQP isoforms for which the functional full-length paralogues are under purifying selection (Appendix A). It is established that the normal process after a duplication of the genome is that some paralogs become silenced and finally become pseudogenes. In the cultivated olive, that would concern 41 duplicate *OleurAQP* gene sequences, presenting deletion of a portion of the duplicate sequences and/or pseudogenization through loss-of-function mutations (Appendix A). As for the rest, that is 27 full-length duplicates, the last duplication is so recent that all duplicated genes coexpressed with their paralogues.

Nonetheless, some nuances in the transcriptional profiles can be highlighted. Several *Oleur*AQP duplicated gene pairs displayed quite similar expression patterns (*Oleur*PIP1;4; *Oleur*PIP1;6; *Oleur*PIP2;2; *Oleur*PIP2;8; *Oleur*TIP1;2; *Oleur*SIP1;1). This would indicate that their biological functions might not be diverged during plant development and stress responses, but determining whether these genes had similar functions in different tested tissues would require further analysis. In contrast, functional differentiations seem to address five duplicated gene pairs which show different expression patterns in a tissue-specific manner or during stress: *OleurPIP2;1ab*, *OleurPI2;6ab*, *OleurPI2;9ab*, *OleurTIP1;3ade* and *OleurTIP2;3ab*. For these paralogues, the putative recent duplication event might have been a way to control specific expression of particular genes according to developmental and environmental conditions. Most probably, the coding regions have gained new regulatory ways by insertion or deletion of tissue-specific enhancers or repressors, causing spatial and temporal variations in expression patterns of duplicated genes [95]. In addition to specific cis-acting elements that would modulate expression of some *Oleur*AQP genes, the different expression patterns between two homologous genes can also be explained by difference in intron lengths. Implication of introns in the regulation gene expressions in different biological contexts is still under-estimated, and yet, although the intron regions are spliced out during transcription, introns have increasingly been shown to serve important biological functions; notably, they can elevate gene expression without functioning as a binding site for transcription factors [96,97]. This phenomenon was termed ‘intron-mediated enhancement’. Hence, it can be speculated that the intron-exon organization of some *Oleur*PIP and *Oleur*TIP can play regulatory roles in olive tree responses to various abiotic stresses and developmental processes. Further studies of the expression patterns of *Oleur*AQP gene family members by including these intron divergences would facilitate a more comprehensive understanding of the specific functions of these genes, and ultimately, it will aid in selecting candidate genes for functional analysis of their role in specific tissues.

However, overall, the expression divergence between some *Oleur*AQP duplicated gene pairs can be considered as an important first step in functional divergence, increasing the likelihood of their retention in the genome after the duplication event. Incidentally, it would also indicate that the *Oleur*AQP duplicates experienced functional diversity during the evolutionary process tied to the domestication of the olive trees, possibly combined with a gain of specific physiological roles during plant development and stress tolerance.

## 3. Materials and Methods

### 3.1. Genome-Wide Identification of Aquaporin Genes in Wild and Cultivated Olive Trees

Aquaporin genes and deduced protein sequences of the wild olive tree *O. europaea* subsp.* europaea* var. *sylvestris* (v1.0) were retrieved from the phytozome database, a Plant Comparative Genomics portal of the Department of Energy’s Joint Genome Institute (https://phytozome.jgi.doe.gov/pz/portal.html#!info?alias=Org_Oeuropaea_er). The investigations were conducted using keyword queries (“Major Intrinsic Protein” and “Aquaporin”), complemented by *t*BLASTn searches with conservative criteria requiring a cut-off of *E*-value of 1.0^−5^ by using as queries the amino acid sequences of aquaporin sequences from *A. thaliana* (*At*AQPs) [35], *Populus trichocarpa* (*Poptr*AQPs) [30] and *Solanum lycopersicum* (*Sl*AQPs) [19], obtained from previous reports. Every putative aquaporin sequence (*Oeu*AQPs) was carefully scrutinized, including expected AQP motifs (NPA and ar/R selectivity filter) and the prediction of the transmembrane topology, with Interproscan from EMBL (http://www.ebi.ac.uk/Tools/pfa/iprscan/) and the NCBI Conserved Domain Database (https://www.ncbi.nlm.nih.gov/Structure/cdd/wrpsb.cgi). The identified sequences of *Oeu*AQP were subsequently employed as queries to recover, using *t*BLASTn searches, their analogous *Oleur*AQPs from the related cultivated olive tree *O. europaea* L. var. Picual available with the OliveTreeDB website (https://genomaolivar.dipujaen.es/db/index.php). This investigation was also complemented with a search by using annotation keyword queries (“Major Intrinsic Protein” and “Aquaporin”). Putative AQP sequences from the Oleaceae related species *Fraxinus excelsior* were retrieved from the Hardwood Genomics Project HWG website (DOI: https://doi.org/10.25504/FAIRsharing.srgkaf) by using *Sl*AQP, *Oeu*AQP and *Oleur*AQP sequences as queries. *F. excelsior* AQP sequences were specifically annotated for this work according to the standardized gene nomenclature which obeys the following format: genus (*three letter*)—species (*two letters*)—gene name (locus) or MIP nomenclature (subfamily x;y). All AQPs from *O.* var. *sylvestris* (*Oeu*AQPs), *O.* cv. Picual (*Oleur*AQPs) and *Fraxinus excelsior* (*Fraex*AQPs) are “roughly” named in the databases of origin. For this study, we renamed each sequence according to the standardized MIP nomenclature. In order to eliminate any ambiguity, each AQP appellation was phylogenetically established to the ones derived from *A. thaliana*, *P. trichocarpa* and *L. esculentum*. *Oeu*AQP and *Oleur*AQP protein names, accession numbers and sequences (genomic, CDS and deduced proteins) used in this work are listed in Figure 3 and Appendix A, respectively. *Fraex*AQP protein sequences are listed in Appendix A, respectively.

### 3.2. Multiple Sequence Alignment, Phylogenetic Analysis and Subfamily Classification

From the sixty-eight candidate AQP sequences recovered, fifty-two full-length deduced peptide sequences were aligned together with orthologous *At*AQPs, *Sl*AQPs and *Poptr*AQPs using ClustalW software (http://www.ebi.ac.uk/Tools/clustalx2/index.html). The results were viewed with Jalview (v2.10.5). A phylogenetic analysis based on the deduced peptide sequences of AQPs encoded by tomato, *A. thaliana* and poplar was used to assign the *Oeu*AQP sequences to the five established AQP subfamilies (i.e., PIPs, TIPs, NIPs, SIPs and XIPs). Functional annotations were initially attributed for most of the *Oeu*AQP loci in the databases of origin and renamed as mentioned above. The phylogenic studies were done with the *AQP*-deduced peptide sequences. A total of 323 amino acid sequences were aligned using the ClustalW progressive alignment method. Then, the phylogenetic tree of the sequence dataset was inferred using the maximum parsimony method. Maximum parsimony analysis was conducted using the subtree-pruning-regrafting (SPR) algorithm and was bootstrapped with 5000 replicates. All analyses were performed using the MEGA X program. A graphical representation and edition of the phylogenetic tree were performed with iTOL software (https://itol.embl.de/) [98].

### 3.3. Chromosomal Location, Gene Structure, Structural Features of OeuAQPs Values

The phytozome assembly database was used to reveal the chromosomal location for each *Oeu*AQP and to determine their related intron/exon structure. The exon–intron organization of *OeuAQP* genes were compared together using Gene Structure Display Server (GSDS) software (http://gsds.cbi.pku.edu.cn) [99] based on the alignment of CDS sequences with the corresponding genomic sequences. Amino acid (AA) translations were obtained using Bioinformatics Resource Portal Expasy software (https://web.expasy.org/translate/). Protein features such as the theoretical isoelectric point (*p*I), the molecular weight (MW) and the grand average of hydropathicity (GRAVY) were calculated using the ProtParam program (https://web.expasy.org/protparam/). The classical conserved domains, i.e., NPA motifs, the ar/R filter, the Froger’s positions and a panel of post-translational and regulatory positions (i.e., phosphorylation, methylation, the leucine blocking residue and the histidine residue involved in gating) were manually identified based on multiple sequence alignments with heterologous *Sl*AQPs and *At*AQPs. The transmembrane regions were predicted using the TMHMM software tool (www.cbs.dtu.dk/services/TMHMM/) [100] and SOSUI tools (http://harrier.nagahama-i-bio.ac.jp/sosui/sosui_submit.html) [101] and, if necessary, manually corrected with heterologous *Sl*AQP and *At*AQP sequences. The subcellular localizations of *Oeu*AQPs were predicted using the online tools WoLF PSORT (http://wolf-psort.hgc.jp) and Plant-mPLoc (http://www.csbio.sjtu.edu.cn/bioinf/plant-multi/) [102]. The conserved motifs were detected using the multiple em for motif elicitation (MEME) software (http://meme-suite.org/tools/meme) [103], with the maximum number of motifs set as 15. Specificity-determining positions (SDP1–9) were collected from multiple sequence alignments, which were performed using ClustalW as described in [21]. Protein sequences corresponding to functional characterized AQPs transporting non-aqua substrates were obtained from Genbank according to related literature. To evaluate the evolution pressure of AQP genes, the *K*a/*K*s ratio was calculated. The non-synonymous (*K*a) and synonymous (*K*s) substitution ratio of AQP genes was calculated to test selection pressure. Transcript sequences and their deduced protein previously aligned with ClustalW (https://www.genome.jp/tools-bin/clustalw) were submitted to the online program PAL2NAL (http://www.bork.embl.de/pal2nal/) [104]. A ratio significantly greater than 1 indicates positive selective pressure. A ratio around 1 indicates either neutral evolution at the protein level or an averaging of sites under positive and negative selective pressures. A ratio less than 1 indicates pressures to conserve protein sequence.

### 3.4. RNA-Seq Analysis

*Oleur*AQP transcriptional steady state levels on plant organ/tissues were recorded from the plant samples that were obtained and studied by [105]. Briefly, samples were collected from roots, stems, meristems, leaves, flowers and fruits (epicarp and mesocarp) of three healthy 10-year-old cv. Picual olive trees under field conditions. Two biological replicates per sample were sequenced, and each biological replicate consisted of an equilibrated pool of three plant RNAs.

Early development plant samples were obtained and studied by [92]. Briefly, seeds of cv. Arbequina were induced to geminate. Seedlings were grown in vitro under chamber conditions with a 16 h photoperiod of fluorescent light and at 25 °C constant temperature until they were two months old and then potted and grown in a conditioned greenhouse (25 °C). The aerial parts of 10 plants were collected at 1, 2, 3, 4, 5 and 6 months after seed activation. Two biological replicates of ten pooled plants per sample were sequenced.

*Verticillium dahliae* infected plant samples were obtained and studied by [87,106]. Briefly, 40 plants were infected by root-dip inoculation in a conidia suspension (10^7^ conidia mL^−1^) of defoliating *V. dahliae* isolate V937I. Forty non-inoculated plants manipulated in the same way were used as a control group of plants in the absence of the pathogen. Three plants were pooled for each biological replicate.

The cold stress was induced and the RNA-seq performed by [88]. Briefly, 35 four-month-old potted olive plants of the cv. Picual acclimated at 24 °C, and then incubated with a 14 h photoperiod of fluorescent light at 65 µmoles m^2^ s (10 °C day/4 °C night) for 10 days and constant 76%–78% relative humidity (RH). An additional group of 15 plants were used as a control treatment. Aerial tissues were then harvested at 0, 24 h and 10 days (three plants/time point). Three plants were pooled for each biological replicate.

Each sample was collected and frozen in liquid nitrogen immediately and stored at −80 °C prior to the molecular assays. The total RNA was extracted using a Spectrum Plant Total RNA Kit (Sigma-Aldrich, St. Louis, MO, USA) according to the manufacturer’s instructions. Two technical replicates of each sample were then sequenced on different lanes in the flow cell using paired-end sequencing (101 × 2) in an Illumina^®^ HiSeq sequencer (Illumina, San Diego, CA, USA).

## 4. Conclusions

To the best of our knowledge, this is the first comprehensive study and systematic genome-wide analysis of AQP gene families in *O. europaea* L. This study highlighted several novel findings explaining the structural conservation and possible functional diversity of AQPs in wild and cultivated olive tree varieties and their involvement in cell responses to various challenged environments. Overall, 68 and 127 non-redundant AQP genes were identified in the wild variety (*Oeu*AQPs) and the cultivated cultivar (*Oleur*AQPs), including 52 and 79 putative functional AQPs, respectively. All *Olea* AQP belong to five distinct subfamilies: PIP, TIP, XIP, NIP and SIP. This doubling observed in the cultivated cultivar could be related to classical polyploidization and a recent gene duplication phenomenon. Despite this expansion, every paralogous full-length AQP pair encodes very similar proteins, and the overall gene structure is highly conserved. This conservation results from a quite recent expansion of the sequences, for which strong purifying selections act on the duplicated gene pairs identified in the domesticated species. Expression profiling of *Oleur*AQP genes revealed high expression of various PIP1s, PIP2s and TIPs, in almost every organ and tissue, suggesting the probable important roles of these AQPs in the early developmental processes and abiotic and biotic stress responses. Because aquaporins play prominent roles in a plethora biological processes, further studies are required to ascertain the functions of the individual selected *OleurAQP* genes identified in this study and to reveal more functional mechanisms for these genes. All these results increase our knowledge of the molecular mechanisms behind the actions of AQPs in olive domestication, and the integration of bioinformatics analysis with biological experiment validations will provide further insight into the key roles that some AQPs play in development processes and stress tolerance in domesticated olive trees.

## Figures and Tables

**Figure 1 ijms-21-04183-f001:**
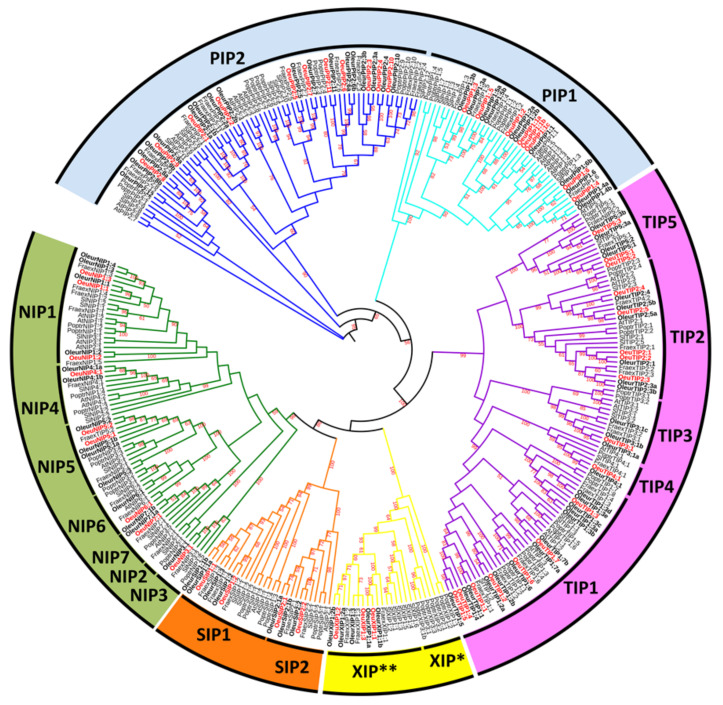
Phylogenetic analysis of protein sequences of the 52 *Oeu*AQPs and the 79 OleurAQPs with *Fraxinus excelsior*, *Arabidopsis thaliana* and *Populus trichocarpa* hortologs. Deduced amino acid sequences were aligned using ClustalW, and the phylogenetic tree was constructed using the maximum parsimony method. Maximum parsimony analysis was conducted using the subtree-pruning-regrafting algorithm. The number next to the branches represents bootstrap values ≥50% based on 5000 resampling. The distance scale denotes the number of amino acid substitutions per site. The AQP sequences from *O. europaea* var. *sylvestris*, *O. europaea* cv. Picual, *F. excelsior*, *A. thaliana* and *P. trichocarpa* are preceded by the prefixes Oeu, Oleur, Fraex, At and Poptr, respectively. *Oeu*AQPs are mentioned in red, and *Oleur*AQP in bold. The name of each subfamily and subgroup is indicated next to the corresponding group. XIP* and XIP** correspond to XIP sequences from Rosids and Asterids, respectively. *Oeu*AQP accession numbers and sequences are listed in Figure 3 and Appendix A*. Oleur*AQP and *Fraex*AQP accession numbers and sequences are listed in Appendix A, respectively. This Figure 1 is also presented in Appendix A with high definition for ease of reading.

**Figure 2 ijms-21-04183-f002:**
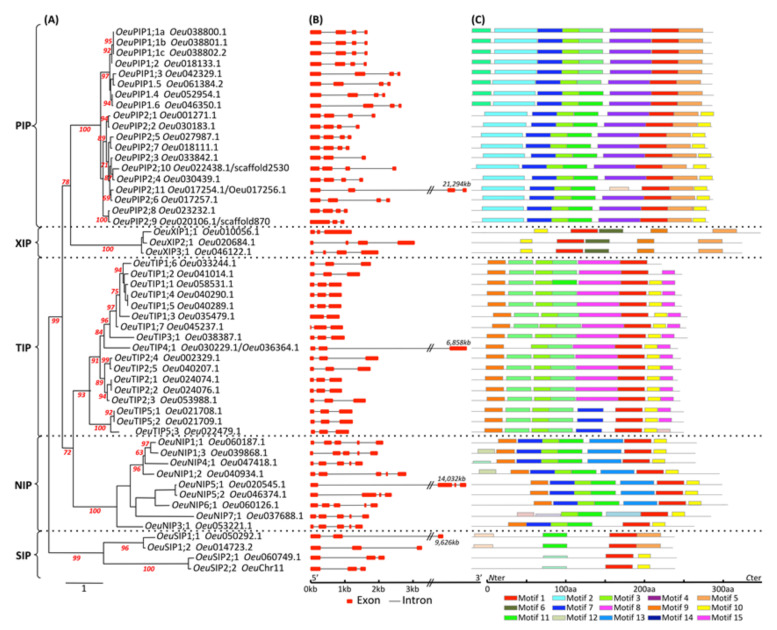
(**A**) Phylogenetic relationship, (**B**) Exon/Intron genomic structure, and (**C**) protein motif organization of the 52 full-length *Olea europaea* var. *sylvestris* aquaporin sequences. (**A**) *Oleu*AQP Sequences were aligned with MUSCLE, and the unrooted phylogenetic tree was reconstructed using the maximum likelihood method. The numbers at the nodes represent the percentage bootstrap values (only those >50% were represented). The reliability for the internal branch was assessed using 1000 bootstrap replicates. The distance scale denotes the number of amino acid substitutions per site. *Oleu*AQPs clustered into five AQP subfamilies: *Oleu*PIPs, *Oleu*XIPs, *Oleu*TIPs, *Oleu*NIPS and *Oleu*SIPs. (**B**) Exons and introns of the *Oleu*AQP genes are represented by red boxes and black lines, respectively. Gene structures were compared using GSDS software. Gene orientations are indicated (5′—3′) in the x-axis. (**C**) Distribution of the conserved motifs among the *Oleu*AQP proteins. Motif analysis was performed by using the MEME web server. Fifteen conserved motifs were identified, and the different motifs are identified using different colored boxes, as indicated at the bottom of the figure. Each color block in the different proteins indicates a specific motif, for which the amino acids are detailed in Appendix A. Protein orientations are indicated (*N*ter-*C*ter) on the x-axis.

**Figure 3 ijms-21-04183-f003:**
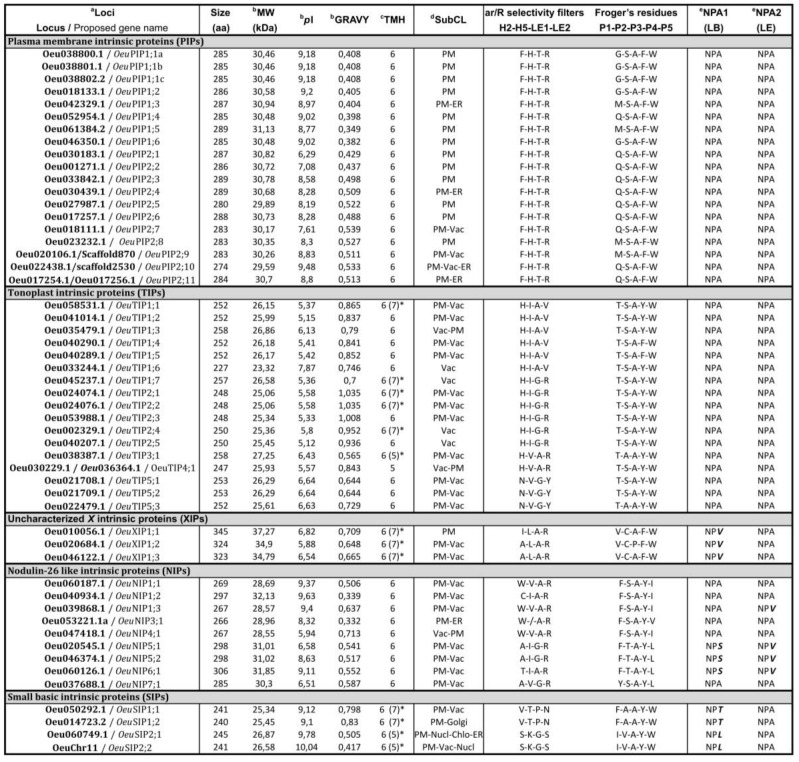
Nomenclature and protein properties of the wild species *Olea europaea* var. *sylvestris* aquaporins (complete sequences). ^a^ Loci, gene IDs and AQP location are based on Phytozome assembly v1.0; ^b^ MW, protein molecular wight; *p*I, protein isoelectric point; GRAVY, ground average of hydropathy; ^c^ TMH, number of transmembrane helices predicted by TMHMM and SOSUI analysis tools; * regions adjusted according to alignment with characterized orthologs from Arabidopsis, poplar and Tomato; ^d^ DubCl, predicted subcellular localization: PM, plasma membrane; Vac, vacuolar; ER, endoplasmic reticulum; Chlo, chloroplast; Nucl, nuclear. ^e^ NPA: Asparagine, Proline, Alanine; Bold italic letters denote usual amino acids in NPA motifs.

**Figure 4 ijms-21-04183-f004:**
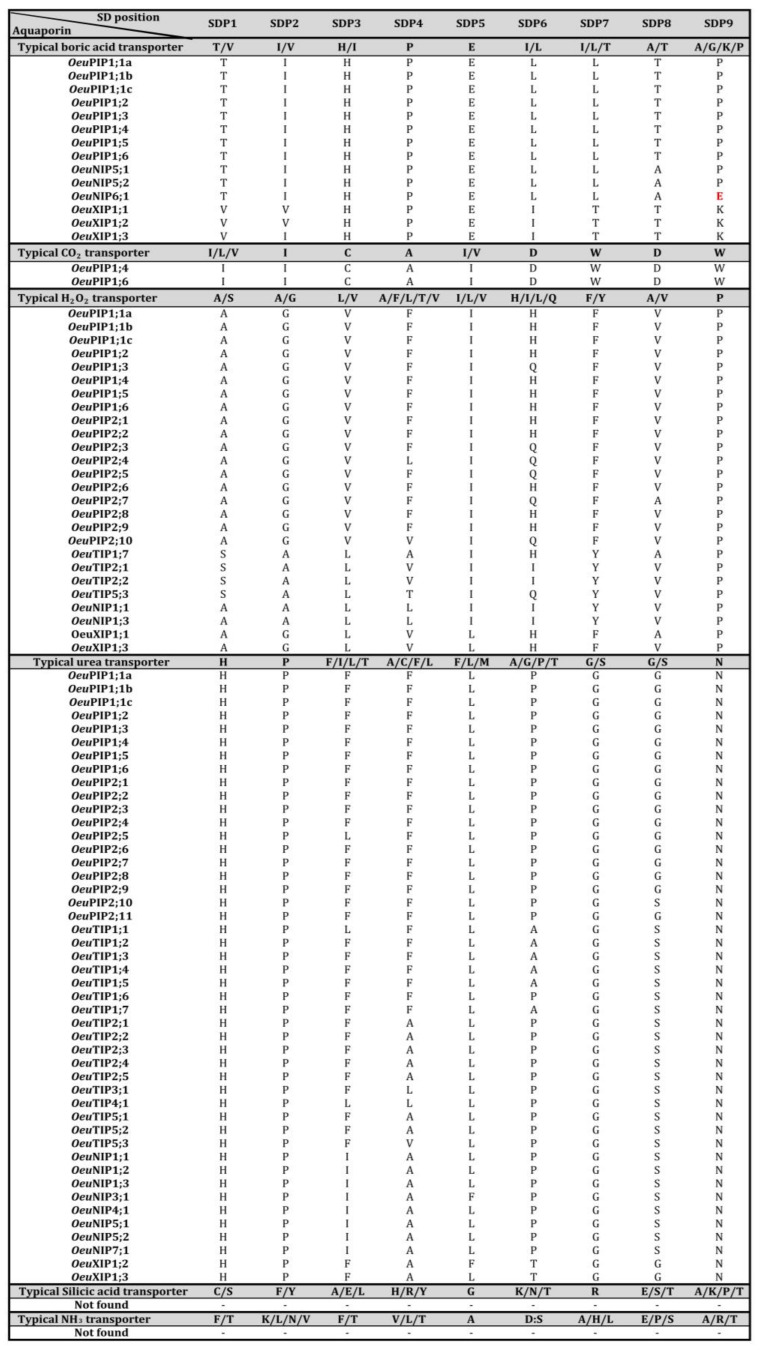
Summary of typical specificity-determining positions (SDPs) and those identified in the complete sequences of *Oeu*AQPs from the wild olive species *Olea europaea* var. *sylvestris*. The red residue represents a putative novel site. Amino acid alignments of the *Oeu*AQPs transporting non-aqua substrates where SDP signatures were identified are shown in Appendix A.

**Figure 5 ijms-21-04183-f005:**
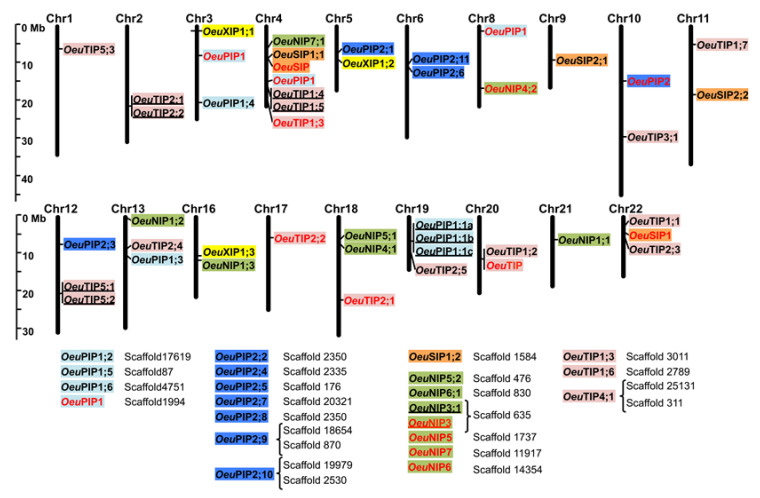
Genomic distribution of *Oeu*AQP genes on the 23 *Olea europaea* var. *sylvestris* chromosomes (except chromosomes 7, 14, 15 and 23). The different aquaporin subclasses are distinguished using different colored boxes that correspond to the color code in Appendix A. The scale is in megabase pairs (Mb). Sixteen sequences could not be anchored to a specific chromosome and are specified in their scaffold of origin. Pseudogenes are specified in red. Tandemly duplicated genes are underlined.

**Figure 6 ijms-21-04183-f006:**
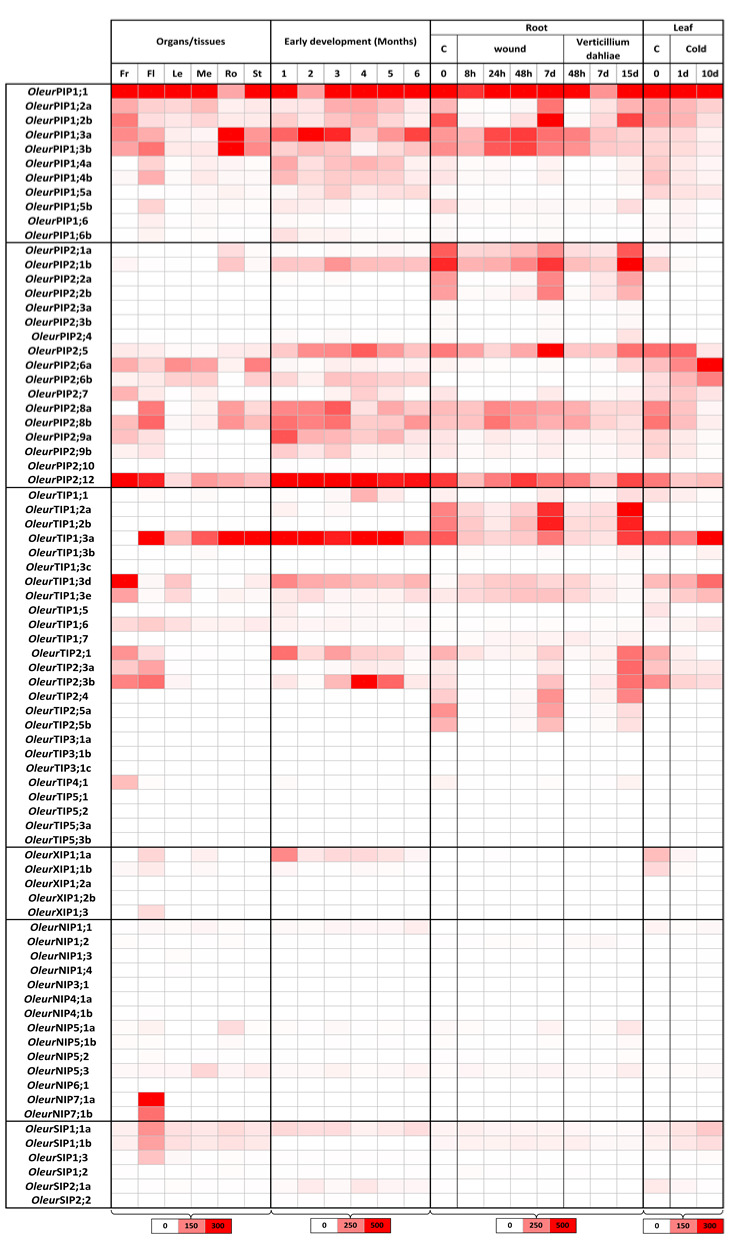
Expression profile of *Oleur*AQP genes from the cultivated *Olea europaea* cv. Picual in different physiological conditions. *Oleur*AQsP are grouped by subfamily. Fr, fruit; Fl, flower; Le, leaf; Me, meristem; Ro, root; St, stem. Control assays are marked C. h, hours post treatment; d, days post treatment. The horizontal color scales at the bottom of the figure represent the normalized expression of the aquaporins in terms of reads per kilobase of transcript per million mapped reads (RPKM). Expression levels are displayed by the depth of red, and the related values are detailed in Appendix A.

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
