# Peer review of "Genome Wild Analysis and Molecular Understanding of the Aquaporin Diversity in Olive Trees (Olea Europaea L.)"

_ijms, 2020, doi:10.3390/ijms21114183_

Round 1

Reviewer 1 Report

I find this work pretty well prepared, language is good, the part containing information about aquaporins sequence analysis is well prepared, however I can find some issues through this paper that should be fixed:

  1. Figures in this work are very bad quality, all of them should be little bigger, especially figure 1, 3, those two are illegible,
  2. I can’t find proper hypothesis,
  3. The work is too descriptive and too long, I would shortened Introduction and Results section a bit,
  4. Results for transcription profiling are little doubtful (Fig. 5), why in control conditions in the roots we can see similar values as for the latest stress conditions. Shouldn’t control values be more similar to those observed after 8h wound treatment? Since in the roots in organs/tissue (column 5) we observe different values, which I presume, those measurements were prepared also in control conditions. Please explain. Moreover, does this results are statistically significant? Why only two biological replicates were used? How the results differed between samples? I think that information might be important for the readers.

Reviewer 2 Report

In the paper in question, Faize et al. carry out a biocomputational analysis of aquaporins (AQPs) in Olive trees. By means of phylogenetic analysis and sequence similarity, authors identify and classify the 52 and 79 full-length AQPs, respectively from wildtype and cultivated olive trees into the five know plant AQP subfamilies. The possible structural features (i.e, pore constrictions, pore selectivity, transmembrane domains) and functions of the Olive tree AQPs are discussed. Most of the information comes from in silico analyses, however, the work in question may reveal a valuable genetic resource for future experimental studies aimed at defining the molecular mechanisms underlying stress tolerance and nutrient uptake in Olive trees. This manuscript informative and interesting. There are some points that need to be addressed.

-OleurPIP1s doesn’t seems to be overexpressed under cold stress and only few PIP2s and TIPs seems to be highly expressed (Line: 813). Please also discuss about NIPs expression in various tissues and stress conditions. Overall, the way the authors presented/discussed the expression data need to be revised more precisely.

-The total number of full-length AQPs identified in the present study is not clear. Contrasting statements have been made in the Abstract and main text and conclusions (Line 192-196, Line 984-985).

 “Among the 79 full-length candidates, 26 sequences are in single copy and 26 AQP-types are duplicated, these last generating 27 full-length sequences and 3 pseudogenes. As a result, 53 full-length AQP-types are identified in the cultivated genome, a similar AQP level was recorded in the wild variety.”

“Overall, 64 and 127 non-redundant AQP genes were identified in the wild variety (OeuAQPs) and the cultivated cultivar (OleurAQPs), including 54 and 78 putative functional AQPs, respectively.”

-Line: 605: “The nine tandemly duplicated genes mostly concern the OeuTIP subfamily, and particularly the OeuTIP1 subgroup.” What are these nine tandemly duplicated genes? Please specify them.

-Figures (especially 1 and 3) are of low resolution and should be replaced with high resolution figures.

-Please provide suitable reference here (line: 66).

-Use of nonscientific terms should be corrected throughout the text (For example; “putative operational candidates”, “68 putative AQP outputs”.

-Figure 3. Legend: expand Oe in “Oe. var sylvestris”.

-Replace “dyctyostelium discoidum” with “Dictyostelium discoideum

-Please correct the typos.

“OeuAQP duplicate has Ka/Ks ratios between 0,001 and 0,2201”

 “OleurNIP7;1b which is under diversifying selection with a ratio of 1,1560)”

 “consistently higher expression, is widely reported [18-83-84].”

-Line 130, “In this respect, gene expression is a major step toward a gene elucidation”

-Line 886: “Lycopersicon esculentum” should be replaced with “Solanum lycopersicum

Round 2

Reviewer 1 Report

Dear Authors and Editors,

Thank You for the response, I still have some thoughts about this manuscript I would like to share before the further processing: 

Ad. 1

I can see that all You did was manual increase the size of the figures without any modification and that didn’t worked well.

In the Figure 1 the font is too small for the printed version, the AQP names and genetic distance are barely visible at 150% magnification. I would try to decrease the size of the outer ring and delete the groups names.

Same situation in the Figure 3. If this is impossible to increase the font size to minimum 4 pt. then I don’t see the point to place this figures in the main manuscript body, but remove them to supplementary files.

Ad. 4d

Thank You for the explanation and the table presenting expression data. I highly recommend to put this table containing expression data for both biological replicates in all measurements into supplementary file. I find this data very significant for the readers.  

Author Response

June, 2nd 2020

Dear Reviewer,

Please be assured that we have met your first comment by more than to increase the “size” of our figures. Exactly, we re-corrected their quality and definition. While care has been taken to ensure a very high image quality, the problems of their edition (in pdf or else post submission) are –sometime- recurrent.

Then, the problems concern the figures 1 and 3.

We redesigned the Figure 1 by deleting the subfamily names as you are proposing. Besides that, this last outer ring is redundant and, therefore, unnecessary. In addition, this figure is provided in Supplemental Figure S4, with a very high definition. Concerning the figure 3, you are absolutely right. Therefore, the figure 3 was removed to supplemental Files (Figure S7).

In addition to the supplemental files, we will also propose to the IJMS Edition Services every figures and tables in high definition.

Regarding your last comment, it is obviously wise to provide such a table to the readers. This table (namely Table S5) includes all the biological replicates for all the biological conditions explored in our work.

We corrected all Figures and Supplemental file names in the main text accordingly.

Thank you for your comments and your help to improve our article.

Sincerely yours,